# Astrocytes amplify neurovascular coupling to sustained activation of neocortex in awake mice

Adam Institoris [1], Milène Vandal[1], Govind Peringod[1], Christy Catalano[1], Cam Ha Tran[1,2], Xinzhu Yu [3,4,5], Frank Visser[1], Cheryl Breiteneder[1], Leonardo Molina[6], Baljit S. Khakh[3,4], Minh Dang Nguyen[6], Roger J. Thompson[7] & Grant R. Gordon [1]✉

Functional hyperemia occurs when enhanced neuronal activity signals to increase local cerebral blood flow (CBF) to satisfy regional energy demand. $Ca^{2+}$ elevation in astrocytes can drive arteriole dilation to increase CBF, yet affirmative evidence for the necessity of astrocytes in functional hyperemia in vivo is lacking. In awake mice, we discovered that functional hyperemia is bimodal with a distinct early and late component whereby arteriole dilation progresses as sensory stimulation is sustained. Clamping astrocyte $Ca^{2+}$ signaling in vivo by expressing a plasma membrane $Ca^{2+}$ ATPase (CalEx) reduces sustained but not brief sensory-evoked arteriole dilation. Elevating astrocyte free $Ca^{2+}$ using chemogenetics selectively augments sustained hyperemia. Antagonizing NMDA-receptors or epoxyeicosatrienoic acid production reduces only the late component of functional hyperemia, leaving brief increases in CBF to sensory stimulation intact. We propose that a fundamental role of astrocyte $Ca^{2+}$ is to amplify functional hyperemia when neuronal activation is prolonged.

Neuronal function requires tight and uninterrupted access to $O_2$ and glucose via the blood supply because neurons have neither sufficient energy stores nor capacity for anaerobic metabolism. Thus, neural processing of sensory, motor, and cognitive information drives increased regional cerebral blood flow (CBF) for seconds to minutes as needed. This phenomenon—functional hyperemia—is essential for brain metabolism and also serves as the basis of the blood oxygen level-dependent (BOLD) signal of functional magnetic resonance imaging (fMRI)[1,2].

Functional hyperemia occurs, in part, by large diameter changes in penetrating arterioles[3–6] as a result of multiple parallel neurovascular coupling mechanisms thought to be governed by neurons, astrocytes, and endothelial cells that actuate vascular smooth muscle. Astrocytes respond to neural activity and their peri-vascular processes/endfeet can release $Ca^{2+}$ dependent messengers to regulate arteriole diameter, both ex vivo (reviewed in refs. 7, 8) and in vivo[9,10] when artificially stimulated. We have previously described that astrocytes sense behavioral-state and vascular signals as brief functional

[1]Hotchkiss Brain Institute, Department of Physiology and Pharmacology, Cumming School of Medicine, University of Calgary, Calgary, AB T2N 4N1, Canada. [2]Department of Physiology and Cell Biology, Reno School of Medicine, University of Nevada, Reno, NV 89557-352, USA. [3]Department of Physiology, David Geffen School of Medicine, University of California Los Angeles, Los Angeles, CA 90095-1751, USA. [4]Department of Neurobiology, David Geffen School of Medicine, University of California Los Angeles, Los Angeles, CA 90095-1751, USA. [5]Department of Molecular and Integrative Physiology, Beckman Institute, University of Illinois Urbana-Champaign, Urbana, IL 61801, USA. [6]Hotchkiss Brain Institute, Department of Clinical Neuroscience, Cumming School of Medicine, University of Calgary, Calgary, AB T2N 4N1, Canada. [7]Hotchkiss Brain Institute, Department of Cell Biology and Anatomy, Cumming School of Medicine, University of Calgary, Calgary, AB T2N 4N1, Canada. ✉e-mail: gordong@ucalgary.ca

hyperemia evolves in awake mice[11], yet their capacity to control CBF to a natural stimulus in vivo has not been demonstrated. It remains possible that astrocytes contribute to functional hyperemia only under distinct durations of neural activity. Indeed, astrocyte $Ca^{2+}$ correlates to CBF when sensory stimulation is prolonged[12]. Under anesthesia, astrocyte $Ca^{2+}$ signals can be suppressed[13], and arteriole/CBF responses to sustained sensory stimulation consist of an initial rise followed by a plateau[14]. The latter has prompted many to solely explore initiating mechanisms or assume that there are no temporally distinct components to functional hyperemia. More recent studies, some conducted in awake animals, display CBF/arteriole response curves that appear bimodal[15–17], yet these temporally distinct phases are rarely mechanistically explored or even described. We hypothesized that delayed astrocyte $Ca^{2+}$ signals (>3 s) occurring in the awake state regulate CBF when sensory activation is sustained, therefore comprising an important and unrecognized mechanism of functional hyperemia[12,18]. Thus far, no causal 'necessity' and 'sufficiency' experiments support this hypothesis. The main framework of astrocyte $Ca^{2+}$ in functional hyperemia in vivo relies on correlational studies[12,18–23], none of which have clearly elucidated the physiological context of astrocytic contributions at arterioles. Though recent work suggests a $Ca^{2+}$ independent astrocytic mechanism in CBF regulation involving $CO_2$[24], the foundational work of this field manipulated and measured astrocyte free $Ca^{2+}$[7,8], highlighting the need to resolve the $Ca^{2+}$ signaling hypothesis.

Using 2-photon fluorescence imaging in awake mice and tools to selectively clamp or augment astrocyte $Ca^{2+}$ in vivo, we discovered that during sustained bouts of sensory activation (1) functional hyperemia increases bimodally and (2) astrocyte $Ca^{2+}$ is causal for augmenting the second phase. These findings are important because impairments in sustained functional hyperemia occur in attention deficit[25], age-related cognitive impairment[26], dementia[27], and stroke[28]. Our results help clarify neuro-glio-vascular coupling and will facilitate future work aimed at understanding how astrocytes impact the (patho)physiology of the cerebrovasculature.

## Results

### Sustained functional hyperemia escalates and is associated with astrocyte $Ca^{2+}$ transients

We investigated the temporal dynamics of penetrating arteriole diameter and astrocyte $Ca^{2+}$ during brief (5 s) and sustained (30 s) whisker stimulation using 2-photon microscopy through an acutely installed cranial window over the barrel cortex of awake mice (Fig. 1a). First, imaging Aldh1l1-Cre/ERT2 × RCL-GCaMP6s mice, arteriole responses to whisker stimulation with air puff elicited rapid arteriole dilation that peaked within the first 5 s of stimulation, followed by progressively increasing diameter only when stimulation was maintained for 30 s revealing a bimodal response (Fig. 1b, c). Interestingly, peak dilations to 1 and 5 s stimulation were not different, but dilation to 30 s stimulation was significantly larger due to the putative secondary phase (Fig. 1d). Astrocyte fine processes, followed by endfeet, had $Ca^{2+}$ signals that both increased after the onset of vasodilation (Fig. 1e, f)[11]. Arteriole responses were similar in c57bl/6 mice imaged through thinned skull (Supplementary Fig. 1a–c), confirming that acute craniotomy did not impair functional hyperemia. We found similar arteriole response dynamics and delayed elevation in $Ca^{2+}$ within endfeet of membrane-tethered GCaMP6 expressing Aldh1l1-Cre/ERT2 x R26-Lck-GCaMP6f mice implanted with a chronic cranial window over an intact dura (>4weeks recovery) (Supplementary Fig. 2a–f). However, in this mouse line, we identified two distinct populations of fine process $Ca^{2+}$ signals: 47% of signals started in the first sec of stimulation (ultrafast)[19,21] before initiation of vasodilation and 45% appeared 3–5 s after stimulation onset (delayed) (Supplementary Fig. 2d, e). Ultrafast $Ca^{2+}$ signals were much smaller in amplitude compared to slower $Ca^{2+}$ signals (~4% vs 48%, Supplementary Fig. 2f).

Astrocytes labeled with Rhod-2/AM in acute cranial windows had $Ca^{2+}$ signals that increased after the onset of vasodilation but that remained elevated until stimulation ended (Supplementary Fig. 3a–c), unlike GCaMP6 generated signals that displayed a shorter duration. In further support of a bimodal arteriole response, imaging vascular smooth muscle cell $Ca^{2+}$ changes in PDGFRß-Cre × RCL-GCaMP6s mice during 30 s whisker stimulation showed two separate peaks of $Ca^{2+}$ drop at ~2.5 s and between 20 and 30 s (Fig. 1g, h). This suggests separate processes mediate smooth muscle relaxation in the early and late phase of sustained functional hyperemia.

### Clamping astrocyte $Ca^{2+}$ reduces the late phase of functional hyperemia

Given the stronger association between astrocyte process/endfoot $Ca^{2+}$ and sustained functional hyperemia, and the appearance of a distinct late component in smooth muscle $Ca^{2+}$ traces, we tested for a causal role of astrocyte $Ca^{2+}$ primarily in the late phase of functional hyperemia. Astrocyte $Ca^{2+}$ transients mediate activity-dependent capillary but not arteriole dilation to 5 s stimulation in acute cortical brain slices[29]. Thus, we tested for a causal relationship between long (30 s) but not short (5 s) stimulation-evoked astrocyte $Ca^{2+}$ and penetrating arteriole dilation in acute rat brain slices, by whole-cell patch infusing the $Ca^{2+}$ chelator BAPTA (10 mM) into a network of peri-arteriolar astrocytes. We clamped intracellular free $Ca^{2+}$ close to the resting concentration in astrocytes (~100 nM)[30,31]. We allowed 15 mins for the internal solution to equilibrate in the astrocyte network[32] and for arteriole tone to stabilize. Rhod-2/AM was used to label large astrocyte and neuronal compartments, as well as the neuropil. Before patching an astrocyte, 30 s of high frequency electrical stimulation elicited $Ca^{2+}$ transients in neurons, the neuropil, astrocyte somata, and endfeet, as well as caused arteriole dilation (Supplementary Fig. 4a–e). This was then repeated when the astrocyte network was filled with either control internal or $Ca^{2+}$ clamp internal solution. We found that only the $Ca^{2+}$ clamp solution reduced arteriole dilation to 30 s afferent stimulation (Supplementary Fig. 4d, e). While the astrocyte network filling itself caused reductions in evoked $Ca^{2+}$ from all cell compartments, comparing to the $Ca^{2+}$ clamp solution revealed that the only effect to explain the loss of dilation was the clamp on astrocyte $Ca^{2+}$ (Supplementary Fig. 4d, e). Importantly, neither clamping astrocyte $Ca^{2+}$, nor the control patch, affected arteriole dilation to 5 s stimulation, and patching itself (in any condition) did not affect baseline arteriole tone after whole-cell equilibration (Supplementary Fig. 5a–d). These data show that longer neural stimulation is required to recruit $Ca^{2+}$ dependent astrocyte contributions to arteriole dilation in neocortical brain slices.

Previous attempts to connect astrocyte $Ca^{2+}$ to functional hyperemia in vivo focused on IP3R2 knockout[33–35], but to our knowledge there have been no in vivo manipulations to clamp astrocyte $Ca^{2+}$ more generally for CBF control; thereby affecting multiple $Ca^{2+}$ dependent pathways. We made use of the astrocyte $Ca^{2+}$ silencing tool CalEx[23]. Here, astrocyte-selective AAV (AAV2/5-gfaABC$_1$D-CalEx-HA) delivers a high-affinity $Ca^{2+}$ ATPase pump (hPMCA2w/b) to the plasma membrane (Fig. 2a, b), which reduces evoked and spontaneous elevations in free $Ca^{2+}$ both in vitro and in vivo[23]. As a control, we made a single point mutation in the CalEx sequence (E457A) to prevent $Ca^{2+}$ transport across the cell membrane[36,37]. We validated both tools in awake c57bl/6 mice via intracortical viral injection followed by a chronic cranial window implantation (Fig. 2c). First, we compared astrocyte AAVs for CalEx and GCaMP6f (AAV2/5-gfaABC$_1$D-GCaMP6f) vs the mutated CalEx control virus and GCaMP6f, by examining startle evoked bulk astrocyte $Ca^{2+}$ signals in the barrel cortex (untrained body air puff, Supplementary Fig. 6a, b). Astrocytes expressing functional CalEx showed markedly reduced startle-evoked $Ca^{2+}$ transients compared to control (Supplementary Fig. 6c, d). Post hoc CalEx expression was confirmed by immunostaining against the reporter hemagglutinin (HA) (Fig. 2d).

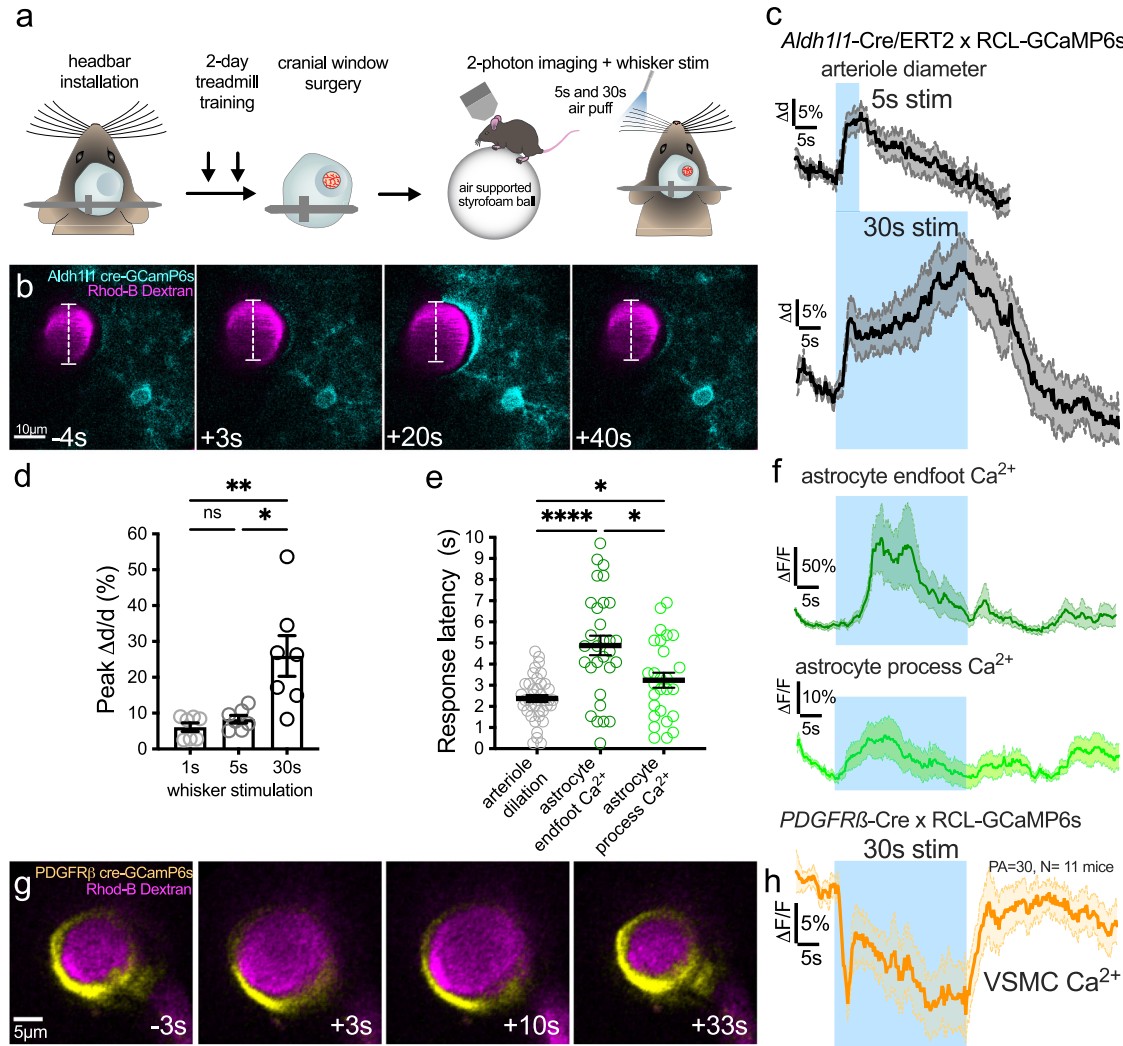

**Fig. 1 | Sustained functional hyperemia escalates and is associated with delayed astrocyte Ca²⁺ transients. a** Cartoon and timeline of the awake mouse, acute cranial window (skull + dura removal), 2-photon imaging experiment in barrel cortex. **b** Time series images showing dilation of a Rhodamine (Rhod)-B-dextran labeled penetrating arteriole (magenta, median filtered image) and Ca²⁺ responses (cyan) of a GCaMP6s-expressing astrocyte in response to 30 s whisker stimulation at 3, 20 and 40 s in an *Aldh1l1*-CreERT2 × RCL-GCaMP6s mouse (representative of *n* = 9 mice). Time stamps refer to stimulation onset as 0 s. **c** Traces of arteriole dilation (black) to 5 and 30 s whisker stimulation. *N* = 28 trials (T) of 12 penetrating arterioles (PA) from 9 mice. **d** Summary data of peak arteriole dilation to 1, 5, and 30 s whisker stimulation. Friedman test (one-sided). *N* = 7 PA from 6 mice. **e** Summary of response onset (calculated from each trial as 3 × SD above baseline) for dilation and astrocyte Ca²⁺ from 13 PA of 9 mice. Arteriole dilation *T* = 39, astrocyte endfoot Ca²⁺:*T* = 30 astrocyte process Ca²⁺: *T* = 27. Mixed effect analysis (two-sided) with Holm–Sidak's multiple comparison. **f** Astrocyte endfoot (dark green) and astrocyte arbor (light green) Ca²⁺ traces to 30 s whisker stimulation. **g** Time series images of GcaMP6s expressing vascular smooth muscle cells (VSMC)(yellow) within a penetrating arteriole loaded with Rhod-B-Dextran (magenta). Each image is an average projection of 4 raw images. **h** VSMC Ca²⁺ trace during 30 s whisker stimulation. *N* = 30 PA from 11 mice. All average trace and summary dot plot data show mean ± SEM. For further statistical details see Supplementary Table 1. Source data are provided as a Source Data file.

In response to 5 s whisker stimulation, we found that astrocyte CalEx had no impact on evoked arteriole dilation compared to control AAV (Fig. 2f, g). Yet remarkably, astrocytic CalEx reduced only the late component (last 5 s of stim) of arteriole dilation to 30 s whisker stimulation compared to control (Fig. 2f, g). These results could not be explained by differences in baseline arteriole tone caused by CalEx expression (Supplementary Fig. 7), and instead suggest that astrocyte Ca²⁺ is important for amplifying arteriole dilation during sustained, but not brief, functional hyperemia.

Confirming a clamp on astrocyte Ca²⁺, CalEx reduced the delayed astrocyte endfoot Ca²⁺ signal occurrence (Fig. 2h) and peak amplitude caused by 5 s (Fig. 2i, j) and 30 s whisker stimulation, as well as the integrated Ca²⁺ response with an overall significance of $P_{CalEx} < 0.0001$ (Fig. 2j). CalEx also diminished astrocyte process Ca²⁺ transient occurrence (Fig. 2k), Ca²⁺ peaks and Area Under the Curve (AUC) Ca²⁺ responses when stimulating for 5 s and for 30 s (Fig. 2l, m). These data

demonstrate that CalEx is an effective tool for clamping astrocyte Ca²⁺ signals in vivo.

It is well established that astrocyte Ca²⁺ signals and subsequent gliotransmission can shape synaptic transmission and neuronal excitability[23,30,38,39]. For example, astrocyte-targeted CalEx in the striatum caused a loss of tonic inhibition in medium spiny inhibitory neurons, yet excitatory inputs onto these cells were unaltered[23]. This prompted us to explore the effect of astrocyte CalEx on neuronal Ca²⁺ signals in the barrel cortex during brief and prolonged whisker stimulation by co-injecting CalEx or its control virus with a pan-neuronal GCaMP6f encoding virus (AAV9.hSyn.GCaMP6f). We recorded bulk neuronal Ca²⁺ signals over a set 5000 μm² area in layer2/3 around a responsive penetrating arteriole (Fig. 2n) and observed that neuronal Ca²⁺ peaked immediately after stimulation onset and gradually declined during 30s stimulation in both control and CalEx mice (Fig. 2o, p). This showed that the overall neural Ca²⁺

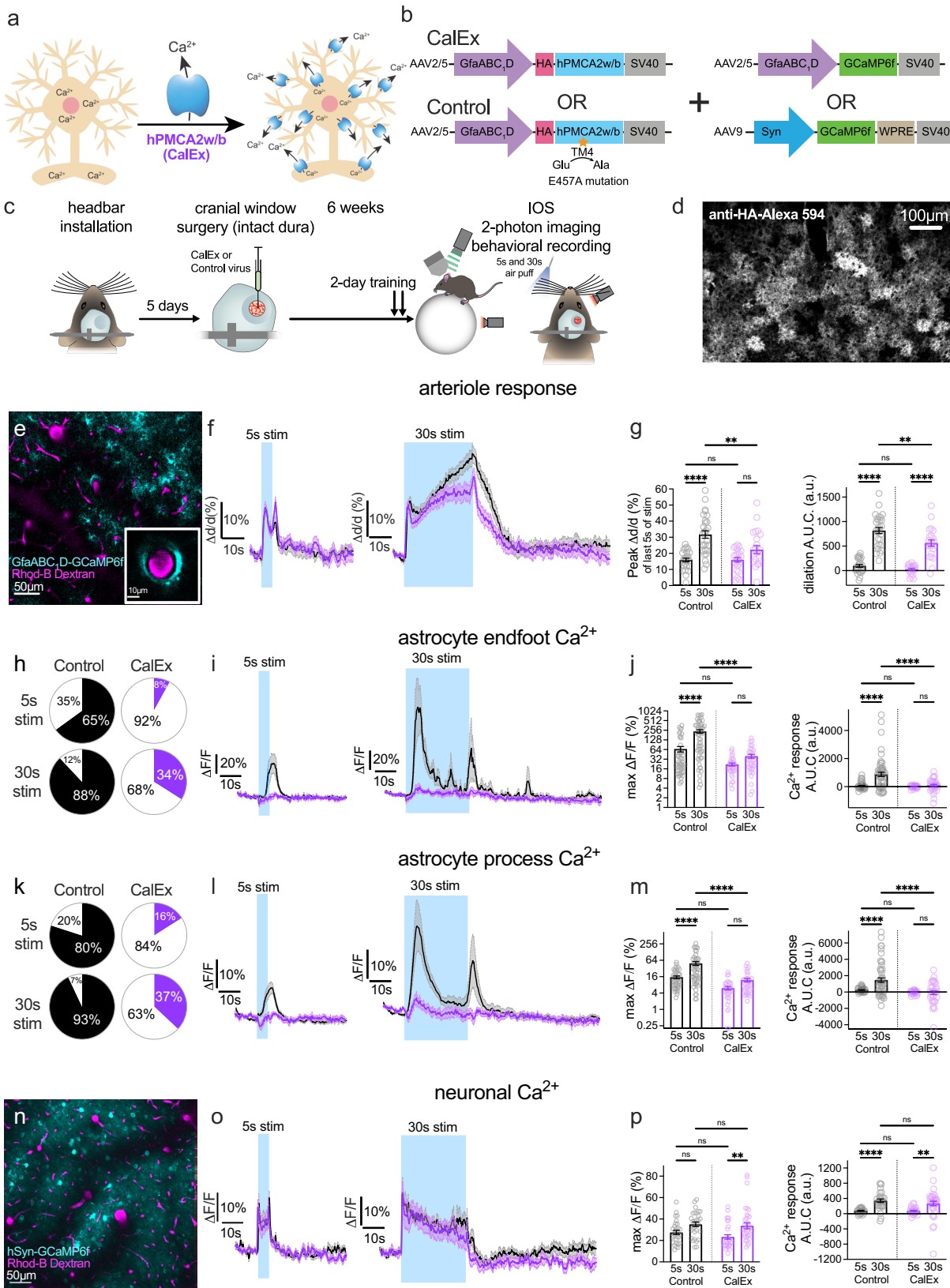

response was unaffected by attenuating astrocyte Ca²⁺ with CalEx. We also performed a more elaborate analysis of discrete Ca²⁺ transients occurring in various neuronal compartments in our field of view using an event-based toolkit[40]. We calculated absolute event frequency, peak Ca²⁺ responses, AUC, absolute spatial area, and duration of many thousands of Ca²⁺ events across the dataset

(Supplementary Fig. 8a–f). Some baseline differences between the groups were noted, which may be due to either a direct effect of astrocyte CalEx on local baseline neuronal activity or small differences in mouse locomotor activity during this time (Supplementary Fig. 8g). The strongest effect of astrocyte CalEx on evoked neuronal Ca²⁺ transients was the reduction in peak amplitudes, which was seen

**Fig. 2 | Clamping astrocyte Ca²⁺ reduces the late phase of functional hyperemia.**
**a** Cartoon of astrocyte Ca²⁺ extrusion tool: a high-affinity plasma membrane Ca²⁺ ATPase hPMCA2w/b (CalEx). **b** Viral vector strategy to express CalEx (*Top Left*) in astrocyte (or control virus- *Bottom Left*) plus GCaMP6f targeting astrocytes (*Top Right*) or neurons (*Bottom Right*). **c** Timeline from viral vector intracortical injection to the imaging experiment. **d** Representative *post hoc* immunofluorescence of CalEx expression against the fused Hemagglutinin (HA) reporter completed in 7 CalEx and 5 Control mice. **e** Representative image of gfaABC₁D-GCaMP6f expressing astrocytes (cyan, median filtered) and Rhodamine (Rhod)-B-Dextran loaded vasculature (magenta, median filtered) during startle in layer 2/3 of the barrel cortex in a control mouse (similar in 5 more mice) (channels were merged from separate acquisitions as startle experiments were performed before vascular dye loading). **f** Average arteriole diameter traces in CalEx and Control for 5 s (*Left*) and 30 s (*Right*) whisker stimulation. **g** *Left:* summary data showing peak arteriole dilation of the last 5 s of the stimulation period. *Right:* net Area Under the Curve (AUC: 40 s from stimulation onset) in the four conditions. Control, 5 s stim: *n* = 26 penetrating arterioles (PA), Control 30 s stim: *n* = 28 PA from 11 mice; CalEx, 5 s and 30 s stim: *n* = 23 PA from 10 mice. **h** Evoked astrocyte endfoot Ca²⁺ event occurrence. Black (control) and purple (CalEx) slices are events, white is no event detected. An event is >3 standard deviation of baseline. **i** Summary time series data

of astrocyte endfoot Ca²⁺ in CalEx vs control for both 5 s (*Left*) and 30 s (*Right*) whisker stimulation. **j** Left: Summary bar graph of peak astrocyte endfoot Ca²⁺ signal. Right: net AUC (stimulation + 10 s) data in the four conditions. Control, 5s stim: *n* = 51 trials, penetrating arterioles (PA), Control 30 s stim: *n* = 57 trials, 18 PA from 6 mice; CalEx, 5 s stim: *n* = 36 trials, and 30 s stim: *n* = 38 trials 12 PA from 5 mice. **k–m** Same as **h–j** but for astrocyte arbor Ca²⁺. Control, 5 s stim: *n* = 48 trials, 16 PA, Control 30 s stim: *n* = 54 trials, 17 PA from 6 mice; CalEx, 5 s stim: *n* = 36 trials, and 30 s stim: *n* = 37 trials 12 PA from 5 mice. **n** AAV9.hSyn.GCaMP6f expressing neurons (cyan) and Rhodamine-B-Dextran loaded vasculature (magenta, median filtered image) in layer 2/3 of the barrel cortex. Representative example of 10 experiments. **o** Average neuronal Ca²⁺ traces in CalEx and Control for 5 s (*Left*) and 30 s (*Right*) whisker stimulation. **p** *Left:* summary data showing peak neuronal Ca²⁺ during the stimulation period. *Right:* net AUC of the stimulation period in the four conditions. Control, 5 s stim: *n* = 30 trials, 10 PA; Control 30 s stim: *n* = 30 trials, 10 PA from 4 mice; CalEx, 5 s stim: *n* = 32 trials, and 30 s stim: *n* = 38 trials 11 PA from 5 mice. All average trace and summary dot plot data show mean ± SEM. All statistical tests are Two-way ANOVA with Tukey's multiple comparisons (two-sided). For further statistical details see Supplementary Table 2. Source data are provided as a Source Data file.

in both 5 and 30 s stimulations (Supplementary Fig. 8b). However, both groups showed no effect on the initial vasodilation early in the response (see Fig. 2f), suggesting this change had no impact on early functional hyperemia. Additionally, during 30 s whisker stimulation, neuronal Ca²⁺ transients exhibited a small increase in AUC (Supplementary Fig. 8c), yet both the AUC and peak effects on neuronal Ca²⁺ signals were observed uniformly across the entire 30 s stimulation period–not selectively on the late phase. Collectively, these data exploring an indirect role for neuronal activity did not adequately explain the preferential effect of astrocyte CalEx on the late component of functional hyperemia.

Cortical astrocytes integrate local sensory information and behavioral state[11,41]. Locomotion, alone[42,43] or concomitantly with tactile stimulation of the whiskers, was shown to trigger astrocyte Ca²⁺ events[42,44], as well as to modulate CBF in the neocortex[17,45]. Notably, two-thirds of barrel cortex excitatory neurons respond predominantly to the combination of vibrissa deflection and running[46]. Therefore, we assessed whether differences in relative locomotion during the stimulation period were confounding the differences in arteriole response and astrocyte Ca²⁺ between CalEx and control mice. Whisker stimulation prompted running at the onset and offset of whisker stimulation (Fig. 3a, b). Mice ran more during and after stimulation than at baseline. We found no difference in average locomotion during the 5 and 30 s stimulation period between CalEx and the control group (Fig. 3a, b). Importantly, arteriole dilation and astrocyte Ca²⁺ signals returned to baseline after the stimulus (see Fig. 2f, i, l). This was despite mice running more after the stimulus than during the pre-stimulus period, showing that running per se did not drive a prolonged increase in arteriole diameter or astrocyte calcium elevation in the barrel. Co-variate analysis of locomotion during the stimulation period in a general linear model confirmed that running does not account for differences in arteriole and neuronal Ca²⁺ responses, as well as in astrocyte process Ca²⁺ caused by CalEx. Interestingly, locomotion influenced the size of endfoot Ca²⁺ signals, but it did not negate the effect of CalEx on endfoot Ca²⁺ transients (statistical analysis in Supplementary Table 3). Collectively, these results suggest that differences in locomotion are unlikely to explain the differences in astrocyte Ca²⁺ and arteriole responses in the late phase of functional hyperemia between CalEx and control.

Arousal provides a robust non-sensory modulation of brain activity across several sensory areas[41,47,48] and has been linked to astrocyte Ca²⁺ transients. The measurement of pupil diameter allows a good estimation of arousal state[49]. We assessed and compared 30 s whisker stimulation-induced pupil diameter changes to a stimulus-free

period and to a startle response elicited by 30 s ipsilateral air puff to the neck in a separate set of trials (Fig. 3c–i). To establish a bidirectionally responsive mid-size pupil, we used ambient green light. In trials with no stimulation, subtle changes in pupil diameter around the baseline corresponded to simultaneous changes in locomotion[50] (Fig. 3c, d). Whisker stimulation for 30 s showed a non-significant pupil dilation that was at least partly reflected by an increase in locomotion (Fig. 3e, g). In contrast, the pupil dilated more robustly in response to startle which was uncoupled from locomotion passed the initial onset period (Fig. 3f, g). These results indicate potentially heightened arousal to whisker air puff which was not as pronounced as a startle reaction. Finally, we confirmed that the pupil reaction in the CalEx group was not different from control (Fig. 3h, i), indicating a similar influence of arousal on functional hyperemia.

## Chemogenetic stimulation of astrocytes enhances the late phase of functional hyperemia

Large astrocyte Ca²⁺ signals occur when synaptic glutamate or neuromodulators stimulate Gq-coupled receptors through inositol triphosphate (IP3) signaling[22,51,52]. However, functional hyperemia is intact in anesthetized, sedated, and awake mice lacking IP3-receptor 2[33–35,53,54], the predominant subtype in astrocytes. Nevertheless, Gq-chemogenetics permits manipulation of cytosolic astrocyte Ca²⁺, which likely impacts signaling cascades outside of IP3R2 as a non-native receptor, allowing us to explore the contribution of general Gq signaling in awake mice. We aimed to test if (1) Gq-DREADD activation in astrocytes directly affects arteriole diameter, and if (2) continuous activation of astrocytic Gq either occludes or amplifies functional hyperemia to short vs long whisker stimulation. We crossed *Aldh1l1*-Cre/ERT2 with CAG-LSL-Gq-DREADD mice to selectively express hM3Dq receptors on astrocytes, which was confirmed by visualizing the fused reporter mCitrine (Fig. 4a). A perforated coverglass was applied to the brain surface to permit stable imaging while allowing topical superfusion of the DREADD agonist compound 21 (C21, 10 μM, Fig. 4b). C21 triggered a prolonged (20–40 min) astrocyte soma Ca²⁺ and endfoot Ca²⁺ elevation, accompanied by transient arteriole dilation (Fig. 4c, d). After an hour of continuous C21, astrocyte Ca²⁺ levels and arteriole diameter returned to baseline (Supplementary Fig. 9). Notably, arteriole dilation to 5 s whisker stimulation remained unchanged in the presence of C21 compared to pre-drug control (Fig. 4e), whereas 30 s stimulation showed a significant augmentation in only the late phase of functional hyperemia in C21 relative to before drug (Fig. 4f). Due to unexplained poor loading of Gq-DREADD positive astrocytes with Rhod-2/AM, we could not

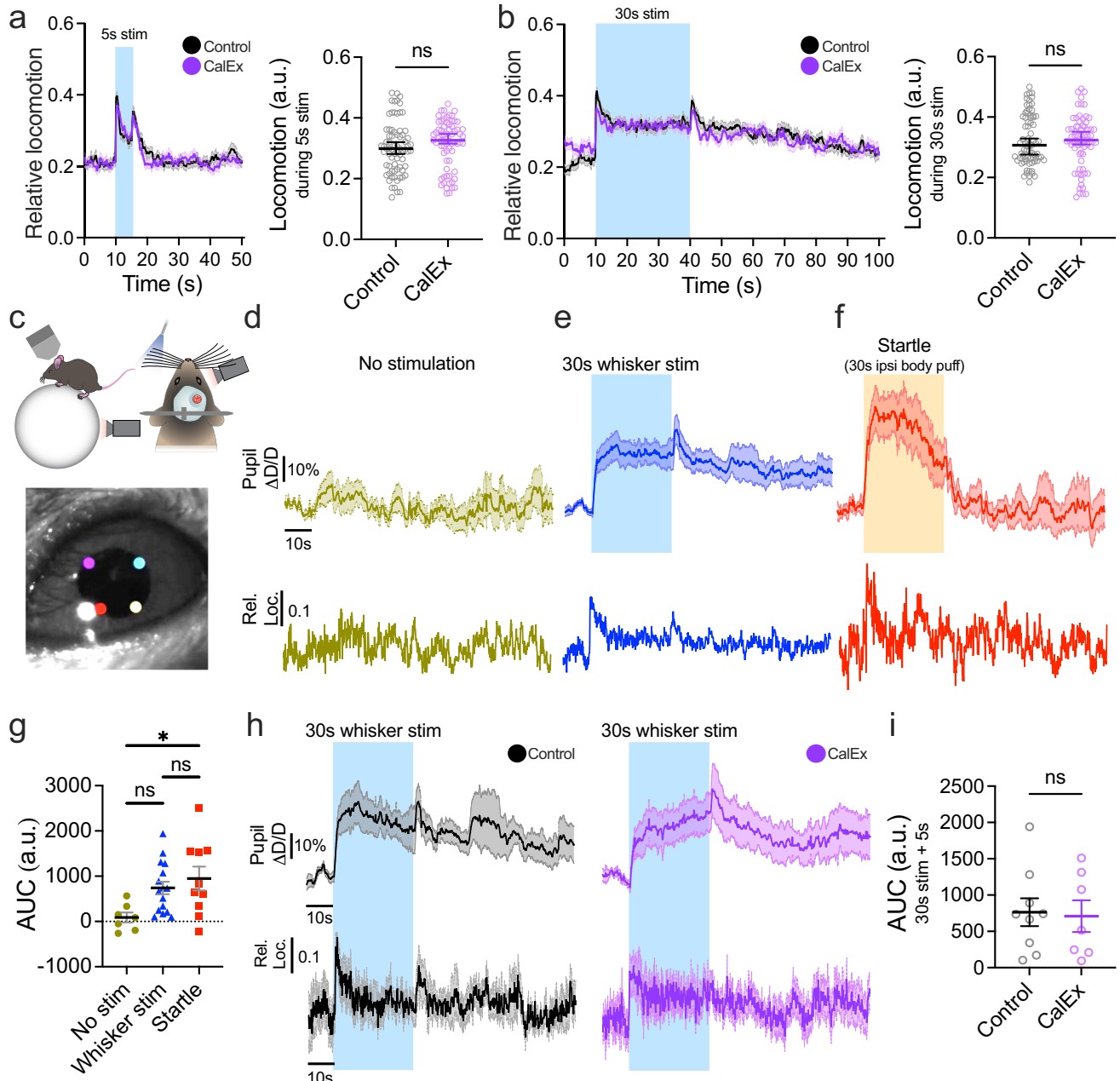

**Fig. 3 | Locomotion and arousal changes do not underlie CalEx effect on sustained functional hyperemia. a** *Left*: Average traces and (*Right*) summary data of relative locomotion during 5 s whisker stimulation for Control and CalEx mice. Mann-Whitney test (two-tailed). Control, 5 s stim: *n* = 73 trials from 11 mice. CalEx, 5 s stim: *n* = 67 trials from 10 mice. **b** Same as *a* but for 30 s whisker stimulation. Control, 30s stim: *n* = 75 trials from 11 mice. CalEx, 30 s stim: *n* = 69 trials from 10 mice. **c** Cartoon of locomotion (*Top Left*) and pupil (*Top Right*) recording with infrared cameras. *Bottom*: Pupil diameter tracking in two axes with the DeepLabCut tracking toolbox performed in *n* = 16 mice. **d** Average traces of relative pupil diameter and relative locomotion (mean only) during no stimulation recording, *n* = 28 trials, 7 mice; during **e** 30 s whisker stimulation, *n* = 68 trials, 16 mice; and during **f** startle evoked by 30 s ipsilateral neck air puff, *n* = 22 trials, 10 mice (**d**–**f**: CalEx and

Control combined). **g** Summary data of net Area Under the Curve (AUC) for pupil diameter for 35 s of no stimulation (*n* = 7 mice), for 30 s whisker stimulation + 5 s post-stimulation (*n* = 16 mice) and 30 s startle + 5 s post-startle period (*n* = 10 mice). One-way ANOVA with Tukey's multiple comparisons. **h** Average traces of relative pupil diameter and relative locomotion during 30 s whisker stimulation in Control (*Left*) and CalEx mice (*Right*). Control: *n* = 38 trials from 9 mice. CalEx: *n* = 30 trials from 7 mice. **i** Summary data of net AUC for relative pupil diameter changes (stimulation + 5 s) in the Control (*n* = 9 mice) and CalEx (*n* = 7 mice) groups. Mann-Whitney test (two-tailed). All average trace and summary dot plot data show mean ± SEM. For further statistical details see Supplementary Table 3. Source data are provided as a Source Data file.

adequately capture endfoot $Ca^{2+}$ signals in response to whisker stimulation before and during C21 application. To test whether the amplification of the late phase of sustained functional hyperemia was indirectly mediated via boosting neuronal activity, another group of *Aldh1l1*-Cre/ERT2 × CAG-LSL-Gq-DREADD mice received an AAV9.hSyn.GCaMP6f injection in the barrel cortex 4 weeks prior to

imaging neuronal $Ca^{2+}$ responses (Fig. 4g). Topical superfusion of C21 for 30 min did not change bulk neuronal GCaMP6 signals at rest (Fig. 4h) and failed to facilitate neuronal $Ca^{2+}$ responses during 30 s whisker stimulation (Fig. 4i). Our results demonstrate that selectively driving the astrocyte Gq-pathway causes arteriole dilation and only influences the late phase of sustained functional hyperemia.

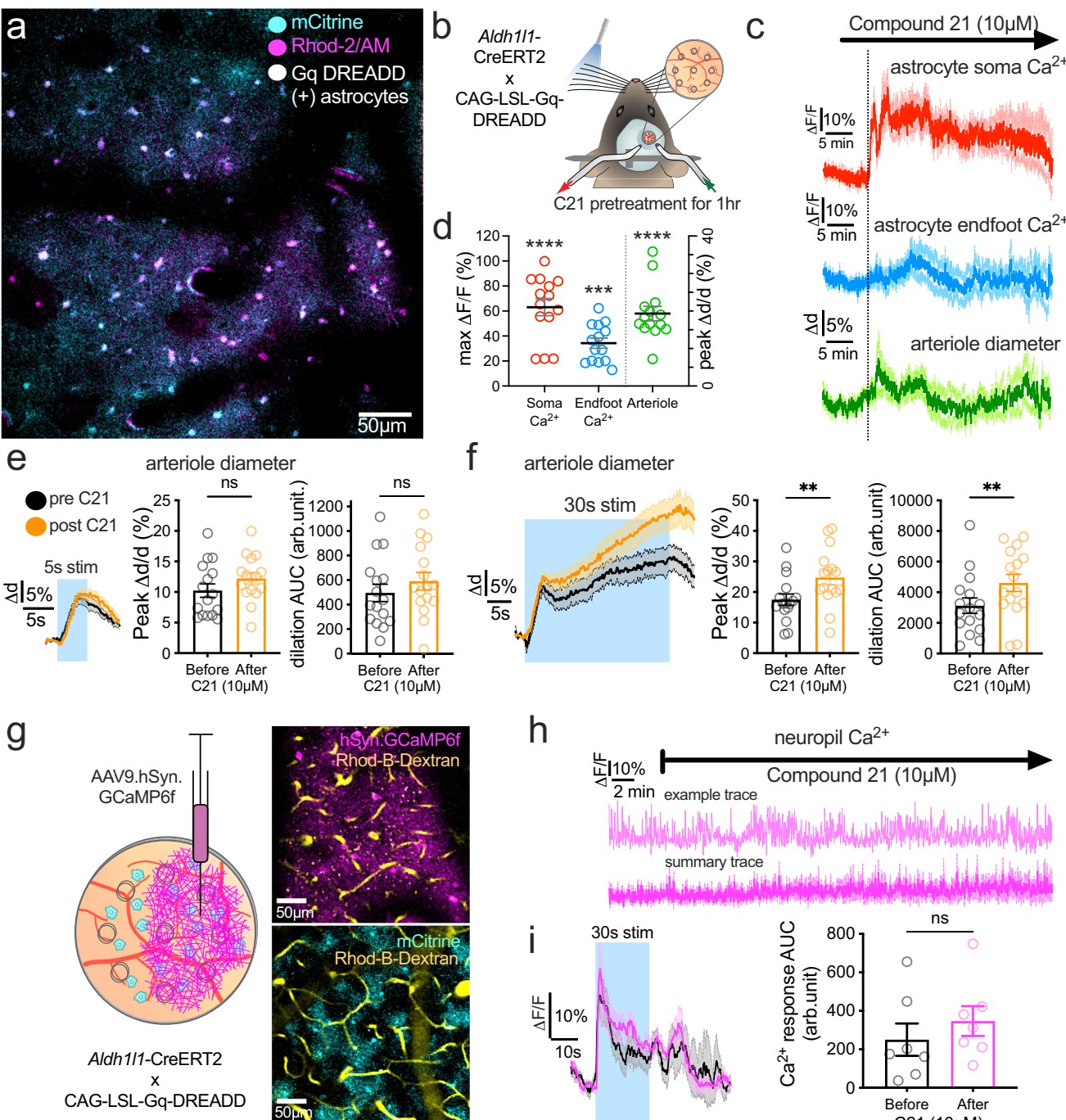

**Fig. 4 | Chemogenetic stimulation of astrocytes enhances the late phase of sustain functional hyperemia. a** Representative 2-photon image in barrel cortex (one from seven animals) of a *Aldh1l1*-Cre x CAG-LSL-Gq-DREADD mouse. Astrocytes are loaded with Rhod-2/AM (magenta, median filtered) and express Gq-DREADD-mCitrine (cyan, median filtered). **b** Cartoon of awake mouse sensory stimulation experiment in the presence of C21 delivered via a perforated window. **c** Average traces of astrocyte soma and endfoot $Ca^{2+}$ and penetrating arteriole (PA) diameter in response to the local superfusion of the DREADD agonist C21 into the perforated window ($n = 7$ mice). **d** Summary data of peak responses to C21 application. All comparisons were Paired $t$ tests (two-sided). Astrocyte soma $Ca^{2+}$: $n = 14$ astrocyte somata and endfeet (average of 38 cells), and $n = 14$ PA. All data are from 7 mice. **e** *Left*: Average arteriole diameter traces in pre-drug control (black) and in the presence of C21 (orange), in response to 5 s whisker stimulation ($n = 16$ PA from 6 mice). *Right*: summary data showing peak dilation and Area Under the Curve (AUC) for 5 s in control and C21. Peak dilation and AUC of dilation: Paired $t$ test (two-sided). $N = 16$ PA, 6 mice. **f** Same as **e** but for 30 s whisker stimulation. **g** *Left*:

Cartoon showing the site of AAV9.hSyn.GCaMP6f injection 4 weeks prior to acute cranial window experiment. *Top Right*: Representative image from experiments in 5 mice of neuronal AAV9.hSyn.GCaMP6f expression (magenta) and Rhodamine-B-Dextran (yellow, median filtered image) labeled vasculature in the lateral side of the window imaged at 920 nm. *Bottom Right*: image of astrocytic mCitrine expression (cyan) in the medial side of the window imaged at 980 nm. **h** Time series measurements of neuronal $Ca^{2+}$ in response to the local superfusion of the DREADD agonist C21 into the perforated window. Upper trace shows a representative region of interest (ROI). Lower trace shows averaged data, $n = 5$ ROI from 5 mice. **i** Average neuronal $Ca^{2+}$ traces in pre-drug control (black) and in the presence of C21 (magenta), in response to 30 s whisker stimulation. $n = 7$ perivascular ROI from 5 mice. *Right*: summary data showing AUC for 30 s stimulation before and after C21. Paired t test (two-sided). All average trace and summary dot plot data show mean ± SEM. For further statistical details see Supplementary Table 4. Source data are provided as a Source Data file.

### NMDA receptors and EETs are involved in sustained functional hyperemia

N-methyl-D-aspartate receptors (NMDAR) are key participants in functional hyperemia. In the canonical model, synapses generate NMDAR-dependent nitric oxide, which causes rapid vasodilation (reviewed in refs. [55–57]). More recent work implicates NMDAR on astrocytes[58] and endothelium[59] in CBF control. Little is known about the contribution of NMDARs to different durations of functional hyperemia in awake animals.

As NMDARs localize to the cell surface, we used *Aldh1l1*-Cre/ERT2 × R26-Lck-GCaMP6f mice (Fig. 5a) to express a plasma membrane-tethered GCaMP in astrocytes and imaged endfeet and processes surrounding a penetrating arteriole (Fig. 5b) or surrounding a precapillary sphincter—a control point for capillary access to blood (Supplementary Fig. 10b, c)[4]. We again applied short (5 s) and long (30 s) duration whisker stimulation before and during topical NMDAR antagonist D,L-AP5 (1 mM) via a perforated coverglass window. The lck-GCaMP6f showed similar astrocyte $Ca^{2+}$ responses to the cytosolic GCaMP6s under control conditions (Fig. 5c, d and see Fig. 1f). Notably, in response to 5 and 30 s whisker stimulation, D,L-AP5 eliminated both astrocyte endfoot and process $Ca^{2+}$ transients around penetrators (Fig. 5c, d), and a similar effect was observed in astrocytes around sphincters but failed to reach significance (Supplementary Fig. 10d, e). In a separate set of experiments using c57bl/6 mice (Fig. 5e), arteriole dilation to 5 s whisker stimulation remained surprisingly unchanged in the presence of D,L-AP5 from control (Fig. 5f). Before NMDAR antagonism, vasodilation was again larger during 30 s stimulation compared to 5 s (Fig. 5f), but in the presence of D,L-AP5, the late component of functional hyperemia was completely blocked, as arteriole diameter returned to baseline quickly after the initial peak (Fig. 5f). These data show NMDARs are critical for astrocyte $Ca^{2+}$ elevation and for the late phase of functional hyperemia.

To test if NMDA inhibition abolished sustained neural activity as a potential explanation for the loss of evoked astrocyte $Ca^{2+}$ elevation and late-phase functional hyperemia, we examined the temporal components of neuronal GCaMP6f during short and long duration whisker stimulation. We used C57BL/6J-Tg(*Thy1*-GCaMP6f) mice to measure neuronal $Ca^{2+}$ signals before and during topical AP5 via a perforated coverglass window (Fig. 5g). We found that AP5 caused a reduction in the peak of the bulk neural $Ca^{2+}$ measurements around the arteriole to 5 s whisker stimulation (Fig. 5h) and also caused uniform reduction in the 30 s response (Fig. 5h). While these data show AP5 effects sustained neural activity, the kinetic profile does not strongly affect the late phase over the early phase, suggesting other NMDAR-dependent neurovascular coupling mechanisms downstream of neural activity were affected, such as astrocytes or endothelium[58–60].

Epoxyeicosatrienoic acids (EETs) have been demonstrated to contribute to functional hyperemia[61–64] and are primarily produced by astrocytes[65,66]. Whether blocking EETs production has differential effects on brief vs sustained functional hyperemia has not been tested in awake mice. We measured functional hyperemia to 5 and 30 s stimulation before and 1 h after i.p. injection of the epoxygenase inhibitor MSPPOH (30 mg/kg) to c57bl/6 mice (Fig. 6a). Consistent with our proposed model, there was no difference in arteriole dilation to 5 s stimulation in the presence of MSPPOH, compared to before (Fig. 6b–d). However, 30 s stimulation was significantly attenuated by MSPPOH relative to control (Fig. 6b–d). MSPPOH yielded a similar result examining short vs long afferent stimulation-induced arteriole dilation in acute cortical brain slices (Supplementary Fig. 11). These experiments show that EETs, likely downstream of NMDAR activation, are important to maintain sustained functional hyperemia.

## Discussion

By selectively attenuating or augmenting astrocyte $Ca^{2+}$ signaling, we provide strong evidence in awake mice that astrocyte $Ca^{2+}$ does not initiate or mediate functional hyperemia when neuronal activation is brief (<5 s) but is essential to amplify the CBF increase when neuronal activation is prolonged. Our work also reveals the origins of the bimodal neurovascular coupling response in vivo. Our data show that a non-neuronal cell type is late to 'come online' and at least partly explain the augmentation of the later phase. Interestingly, earlier work in awake animals showed that surface pial artery dilation to long stimulation was bimodal but the later phase was not amplified[15,16]. As astrocytes do not directly influence surface pial arterioles, this is congruent with only observing amplification of sustained CBF increases at penetrating arterioles which are wrapped by astrocyte endfeet, and that this astrocyte-mediated response is not conducted upstream to the brain surface.

Previously, controversies about the role of astrocytes in functional hyperemia have arisen from challenges in achieving selective activation[67], lack of effect manipulating astrocyte IP3 signaling[33–35,53,54], and side effects of anesthesia/sedation[13,68]. Increased neural activity is the main driver[69] and user of fresh $O_2$[70], yet the linear correlation of BOLD signal to neural activity is lost with sustained brain activation[71,72]. These results are consistent with a large body of literature showing delayed elevations in astrocyte $Ca^{2+}$ when neural activity increases, as well as delayed increases in astrocyte metabolism[73,74]. Our data show that the fully awake animal is necessary to observe sufficiency effects on arteriole diameter driven by astrocyte Gq[33], and suggest other IP3R subtypes play a role in CBF regulation. That selectively driving the astrocyte Gq pathway augmented only sustained functional hyperemia, which is impaired in a number of neurological conditions[25–28], suggests a potential therapeutic target to improve brain blood flow in these states.

Our ex vivo BAPTA and in vivo astrocyte membrane-tethered GCaMP6 transgenic mouse results, as well as CalEx data indicate that ultrafast astrocyte $Ca^{2+}$ signals that precede vasodilation[19,21] are not involved in the initiation or first dilation peak of functional hyperemia. These same data argue against fast acting astrocyte $Ca^{2+}$ at the capillary, pre-capillary, or sphincter level[4,75,76] driving a rapid conduction-mediated dilation to upstream arterioles[77,78]. This is because both astrocyte CalEx in vivo and astrocyte BAPTA ex vivo reached these vascular compartments. While indirect effects of astrocyte $Ca^{2+}$ clamp on neural activity could explain the decrease in sustained functional hyperemia[39,79], that 5 s evoked vasodilation was unaffected both in vivo and ex vivo, and that bulk neuronal $Ca^{2+}$ measures directly around the arteriole in vivo were unchanged by CalEx, argues against this possibility. Furthermore, we found no impact of astrocyte $Ca^{2+}$ clamp on evoked neural $Ca^{2+}$ to long stimulation in brain slices, despite reduced vasodilation. Finally, a selective effect of the EETs synthesis blocker MSPPOH on the late phase, a putative NVC astrocyte pathway, further supports a direct role of endfeet to amplify late phase arteriole dilation. However, it was expected that astrocyte CalEx would have some impact on the local neural activity given the literature[23,39] and because CalEx targets to both perisynaptic processes as well as endfeet. Indeed, our detailed analysis of neuronal $Ca^{2+}$ transients in the parenchyma showed some effects, yet without a preference for the late phase of 30 s stimulation. In either scenario (direct effect on the vessel or indirect effect via neural activity), our data points to an important role for astrocyte $Ca^{2+}$ in this amplification phenomenon.

Notably, the profiles of astrocytic $Ca^{2+}$ responses recorded using Rhod-2/AM and GCaMP6 were different. Rhod-2/AM recorded $Ca^{2+}$ elevations were largely sustained during 30 s stimulation and returned to baseline only after stimulation. Signals reported by GCaMP6 recovered well before the termination of the stimulus. One possible explanation is that GFP-based GCaMP sensors are sensitive to pH[80] and

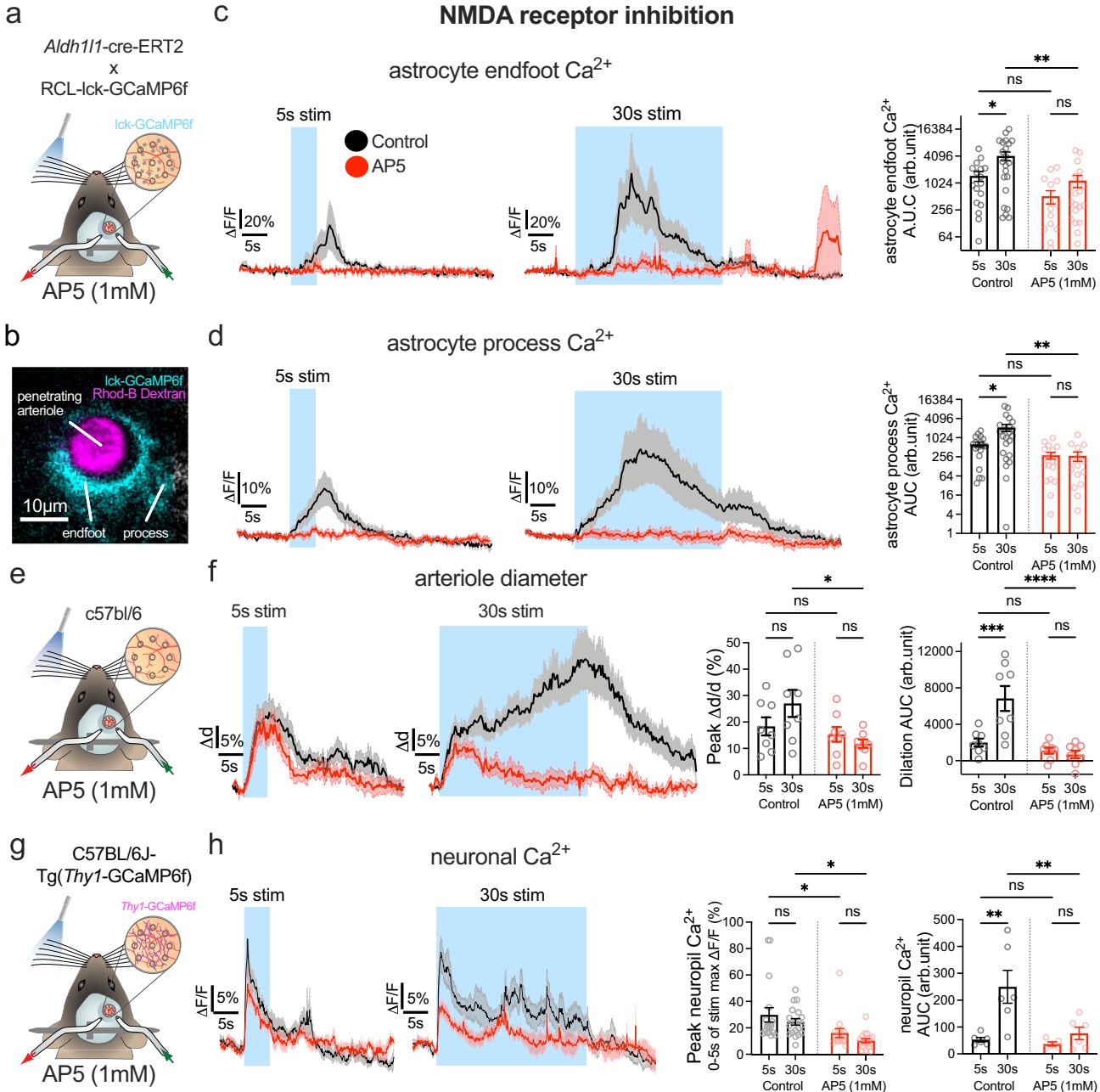

**Fig. 5 | NMDA receptor inhibition abolishes astrocyte activation and reduces the late component of sustained functional hyperemia. a** Cartoon of awake mouse 2-photon imaging experiment with sensory stimulation and perforated window for AP5 superfusion in *Aldh1l1*-CreERT2 × R26-lck-GCaMP6f mice. **b** 2-photon image of a penetrating arteriole (PA) (magenta, median filtered image) and surrounding astrocyte expressing membrane targeted lck-GCaMP6f (cyan). One example from 8 experiments from 8 mice. **c** *Left*: Average time series trace data of astrocyte endfoot Ca²⁺ in pre-drug control (black) and in the presence of AP5 (red) surrounding a PA in response to 5 s (n = 18 trials at 7 PA from 7 mice) and 30 s (n = 25 trials at 8 PA from 8 mice) whisker stimulation. *Right*: summary data of Area Under the Curve (AUC). **d** Same as for **c** but for astrocyte arbor Ca²⁺. 5 s stimulation: n = 18 trials at 7 PA from 7 mice 30 s stimulation: n = 24 trials at 8 PA from 8 mice. **e** Cartoon of a C57Bl/6 mouse with whisker puffer and perforated cranial window for superfusion of AP5. **f** *Left*: Average traces of penetrating arteriole diameter in pre drug control (black) or in the presence of AP5 (red), in response to 5 s or 30 s (n = 8 PA from 7 mice) whisker stimulation. *Right*: summary data of dilation peak and AUC (average of 3–4 trials per vessel). Peak dilation and AUC. **g** Cartoon of a Thy1-GCaMP6f mouse with whisker air puffer and perforated cranial window for superfusion of AP5. **h** *Left*: Average traces of neuronal Ca²⁺ in pre drug control (black) or in the presence of AP5 (red), in response to 5 s (control n = 18 trials, AP5 n = 15 trials) or 30 s whisker stimulation (control n = 19 trials, AP5 n = 18 trials) recorded from 6 perivascular ROIs from 6 mice. *Right*: summary data of maximum neuronal Ca²⁺ signal in the first 5 s of stimulation and neuropil Ca²⁺ signal AUC of stimulation period. Max ΔF/F and AUC. Control 5 s and 30 s: n = 6 ROIs, AP5 5 s: n = 5 ROIs; 30 s: n = 6 ROIs. All average trace and summary dot plot data show mean ± SEM. All statistical tests were Two-way ANOVA with Tukey's multiple comparisons (two-sided). For further statistical details see Supplementary Table 5. Source data are provided as a Source Data file.

## Cytochrome P450 epoxygenase inhibition

**Fig. 6 | Epoxyeicosatrienoic acids explain amplification of sustained functional hyperemia. a** Cartoon of a c57Bl/6 mouse with a sealed cranial window and a whisker air puffer receiving the epoxygenase inhibitor MSPPOH i.p. **b** Average traces of penetrating arteriole (PA) diameter in pre drug control (black) or after MSPPOH treatment (green), in response to 5 s (*Left*) or 30 s (*Right*) whisker stimulation. $N = 10$ PA from 4 mice. **c** Summary data of dilation peak (average of 3–4 trials per vessel). $N = 10$ PA from 4 mice. **d** Summary data of dilation AUC. $N = 10$ PA from 4 mice. All average trace and summary dot plot data show mean ± SEM. All statistical tests were Two-way ANOVA with Tukey's multiple comparisons (two-sided). For further statistical details see Supplementary Table 6. Source data are provided as a Source Data file.

activity-dependent acidification of astrocytes may have led to the quenching of GCaMP fluorescence during the late phase of 30 s stimulation. Alternatively, the difference could be explained by some preferential loading of Rhod-2/AM into mitochondria, an organelle in which GCaMP is excluded. Future experiments should explore whether astrocytic mitochondrial $Ca^{2+}$ dynamics contribute to sustained functional hyperemia. We also acknowledge that astrocyte endfeet and process $Ca^{2+}$ signals recorded in a chronic, fully sealed cranial window are different to that recorded in an acutely implanted, sealed or perforated cranial window. For example, the former additionally revealed elevations that were time locked to the onset and offset of whisker stimulation. While this may reflect responses to onset and offset neurons in the barrel cortex[46], they could also be explained by astrocyte mechanosensing[81] from the blood flow response itself[11,82], which would not be observed in an acute window if the integrity of the intracranial pressure is compromised.

Whisker stimulation elevated locomotion and heightened arousal in our experimental paradigm. These behavioral states are often linked[48], and activating neuromodulatory pathways can facilitate astrocyte activation in the neocortex[42,43]. Two of our results suggest that neuromodulatory systems contribute to functional hyperemia directly through astrocytes. First, the level of locomotion impacted only astrocyte endfoot $Ca^{2+}$ transients and clamping these transients by CalEx significantly reduced late hyperemia[83]. Second, selective activation of Gq signaling, a common pathway for neuromodulators in astrocytes, amplified late hyperemia without modifying bulk neuronal $Ca^{2+}$ responses. Identifying the link between behavior-related

neuromodulatory inputs and astrocyte-derived vasoactive mediators is warranted.

Surprisingly, we found a temporally distinct contribution of NMDA receptors to eliciting delayed astrocyte $Ca^{2+}$ transients and mediating the late phase of functional hyperemia. NMDA receptor activation likely controls astrocyte $Ca^{2+}$ indirectly through diffusible mediators such as nitric oxide released from post-synaptic neurons, because astrocyte-patch based NMDA inhibition with intracellular MK-801 preserves evoked $Ca^{2+}$ transients [58]. Nevertheless, since the block of late phase vasodilation by AP5 is much more pronounced than by astrocyte CalEx, and because AP5 caused a reduction in neuronal activity measured by GCaMP, part of the decrement in the late component of functional hyperemia may be attributed to decreased neuronal activity. It is interesting to speculate that the same synaptic NMDA receptors necessary for long-term potentiation and thus memory are the same for late phase hyperemia, which could have arisen due to the enhanced energy requirements of plasticity[84,85]. However, late-phase hyperemia could also develop by NMDA receptor activation on endothelial cells. This process requires $Ca^{2+}$-dependent D-serine release by astrocytes[59,86] which would be additionally inhibited by AP5 because this treatment abolished sensory-evoked astrocyte $Ca^{2+}$ elevations.

In sum, these data (1) help clarify the literature on the role of astrocytes in functional hyperemia—a phenomenon that is fundamental to fueling the brain; (2) provide important insights into the cell-type specific underpinnings of bimodal neurovascular coupling in awake animals and (3) improve our understanding and interpretation of fMRI data which routinely employs stimulations or tasks of long duration.

## Methods

### Animals

All animal procedures were performed in accordance with guidelines approved by the Animal Care and Use committee of the University of Calgary (protocols AC19-0170 and AC19-0109). Animals were kept on a normal 12–12 h light and dark cycle (7 am: on, 7 pm: off). Room temperature and humidity were set to 21.4 °C and 20%, respectively. Animals had access to food and water ad libitum. Animals were group housed until head-bar installation. Experiments were performed on male P22–90 c57bl/6 mice ($N = 41$: 5 acute window; 11 CalEx; 11 mutant CalEx control, 4 MSPPOH; 7 AP5; 3 thinned skull), transgenic *Aldh1l1*-Cre/ERT2 × RCL-GCaMP6s (Ai96)($N = 9$), *Aldh1l1*-Cre/ERT2 × R26-Lck-GCaMP6f ($N = 13$), *PDGFRβ*-Cre × RCL-GCaMP6s ($N = 11$), *Aldh1l1*-Cre/ERT2 × CAG-LSL-Gq-DREADD ($N = 11$), C57BL/6J-Tg(*Thy1*-GCaMP6f) ($N = 6$). For the use of the Ai96 conditional line, three consecutive tamoxifen injections were administered (100 mg/kg, 10 mg/mL corn oil stock, Sigma, St. Louise MO USA) between the age of P19 to P45 and the awake imaging protocol followed at least 2 weeks later. The other ERT2 expressing mice received 4-hydroxy-tamoxifen for 2 consecutive days between P2–4 (16 mg/kg, 4mg/ml corn oil + 12 v/v (%) absolute ethanol) i.p. in a volume of 20 μl.

### Two-photon imaging

Two custom-built two-photon laser-scanning microscopes were used, one of which was optimized for acute brain slices and patch-clamp electrophysiology[87] and another that was designed for awake in vivo experiments[11]. Both microscopes were supplied by the same Ti:Sapph laser (Coherent Ultra II, 4 W average power, 670–1080 nm, ~80 MHz) and equipped with the following parts: objectives (Zeiss 40× NA 1.0, Nikon 16× NA 0.8), a green bandpass emission filter (525–40 nm) and an orange/red bandpass emission filter (605–70 nm) coupled to photomultiplier tubes (GaAsP Hamamatsu). The open-source scanning microscope software ScanImage (versions 2.9 and 3.81 HHMI/Janelia Farms) running in Matlab[88] was used for image acquisition. During acute brain slice experiments, we acquired time-series images at

0.98 Hz using bidirectional scanning of a 512 × 512 pixel-size area on a single focal plane visualizing fluorescent $Ca^{2+}$ indicator-labeled cells and the center of a penetrating arteriole lumen in a depth of 40–120 μm. For stimulation experiments, the Ti:Sapph was tuned to 850nm to excite Rhod-2/AM and FITC dextran fluorescent indicators. Filling of the astrocyte network with Alexa-488 was imaged at 780 nm.

In vivo experiments using GCaMP6 signals were imaged at 920–940 nm, mCitrine at 980 nm. Imaging astrocytes and vessels was performed at a standard 3.91 Hz imaging frequency, neuronal $Ca^{2+}$ in the CalEx and GqDREADD experiments at 7.82 Hz and in the AP5 experiments at 15.64 Hz. Fluorescent signals were recorded at 50–250 μm depth of the sensory cortex.

## Acute brain slice preparation
We used Sprague-Dawley rats (p21–33) to prepare acute coronal slices of the sensory-motor cortex. Rats were anesthetized with isoflurane (5% induction, 2% maintenance). Two min after a fluorescent dye injection in the tail vein, animals were decapitated and their brains were quickly extracted and submerged into an ice-cold slicing solution (composition in mM: 119.9 N- methyl D-glucamine, 2.5 KCl, 25 $NaHCO_3$, 1.0 $CaCl_2$-$2H_2O$, 6.9 $MgCl_2$- $6H_2O$, 1.4 $NaH_2PO_4$-$H_2O$, and 20 glucose), continuously bubbled with carbogen (95% $O_2$, 5% $CO_2$) for 2 min. We cut 400 μm thick coronal slices with a vibratome (Leica VT 1200S) and placed them to recover for 45 min on a fine mesh in artificial cerebrospinal fluid continuously bubbled in carbogen at 34 °C. ACSF contained the following (in mM): 126 NaCl, 2.5 KCl, 25 $NaHCO_3$, 1.3 $CaCl_2$, 1.2 $MgCl_2$, 1.25 $NaH_2PO_4$, and 10 glucose.

The aCSF was bubbled with a gas mixture containing 30% $O_2$, 5% $CO_2$ and balanced $N_2$ following the recovery for the dye loading period and during imaging experiments in order to enhance vascular dilation responses as described previously[89]. We performed imaging at room temperature (22 °C) and the imaging bath was perfused with aCSF at 2 ml/min by a sealed, gas pressure driven system. The superfused aCSF was supplemented with the arteriole pre-constrictor U46619 (125 nM) starting at least 30 min prior to each experiment to allow for the vessels to develop a steady tone. Drugs were infused into the main superfusion aCSF line (moving at 2 mL/min) through a side line driven by a syringe pump at a rate of 0.2 mL/min at ten-fold of the desired concentration to achieve the intended final concentration in the bath.

## Dye loading
We labeled penetrating arterioles with Fluorescein Isothiocyanate (FITC) dextran (2000 KDa, Sigma Aldrich) via a tail vein injection (15 mg dissolved in 300 μl lactated Ringer's solution (5%)) under anesthesia immediately before slicing. Following slice recovery, slices were bulk loaded in Rhodamine-2 Acetoxymethyl Ester (Rhod-2/AM) (15 μM, dissolved in 0.2% DMSO; 0.006% Pluronic Acid; 0.0002% Cremophore EL) in 3 mL of aCSF while bubbled with a 30% $O_2$-containing gas mixture (see above) in a small incubation well for 45 min at 33 °C. In acute slice experiments astrocytes were identified by a brighter baseline dye fluorescence than neurons due to their preferential Rhod-2/AM uptake and by the presence of endfeet on vasculature.

## Materials
FITC-dextran (2000KDa), rhodamine B isothiocyanate-dextran (Rhodamine-B-dextran, 70KDa), dexamethasone 21-phosphate disodium, Chremophore® EL, DMSO were obtained from Sigma Aldrich. D-(-)-2-Amino-5-phosphonopentanoic acid (AP5) and Compound 21 dihydrochloride was purchased from Hello Bio Inc. (Princeton, NJ). U46619, N-(methylsulfonyl)-2-(2-propynyloxy)-benzenehexanamide (MSPPOH), medotomidine hydrochloride, and atipamezole hydrochloride were ordered from Cayman Chemicals. BAPTA-tetrapotassium and Alexa 488 hydrazide were obtained from Invitrogen, isoflurane from Fresenius Kabi Canada Ltd., Pluronic F127 and Rhod-2/AM from Biotium, buprenorphine from Champion Alstoe

Animal Health, and enrofloxacin from Bayer, meloxicam from Boehringer Ingelheim, ketamine from Vétoquinol N.-A. Inc. AAV2/5.GfaABC1D.GCaMP6f and AAV9.hSyn.GCaMP6f were ordered from Penn Vector Core, PA, USA, pZac2.1-GfaABC1D-mCherry-hPMCA2w/b plasmid was ordered from Addgene.

## Electrical stimulation
The activation of afferent fibers was elicited by electric stimulation using an assembly of a Grass S88X stimulator, a voltage isolation unit and a concentric bipolar electrode (FHC). We softly positioned the electrode onto the surface of the slice by a micro-manipulator ~250 μm lateral to the penetrating arteriole. The tissue was stimulated for 5 s or 30 s with a theta burst pattern (trains of 1 ms mono-polar pulses at 100 Hz for 50 ms, repeated at 4 Hz) using a 0.1 V higher intensity than the threshold of astrocyte endfoot activation (1.1–1.3 V). This threshold was determined by gradually ramping the voltage (+0.1 V) of a low frequency (20 Hz) 5 s stimulation until an endfoot $Ca^{2+}$ transient was observed. For some experiments (see Supplementary Fig. 11), we employed a 20 Hz stimulation for 5 s with low intensity that did not activate astrocyte endfeet and subsequent 30 s stimulations used the same voltage.

## Astrocyte patch clamping experiments
Patching experiments were conducted similar to our previous publication[32]. Before the patching process, slices were electrically stimulated for 5 or 30 s with a theta burst pattern. Next, astrocytes were selected in the 30–70 μm proximity of an arteriole at a 30–60 μm depth, typically above the imaging plane of the arteriole (Supplementary Fig. 4a). A 3–5 MΩ resistance glass electrode formed a GΩ seal on the cell body membrane and when whole cell configuration was achieved astrocytes were filled with an internal solution containing the fluorescent dye Alexa-488 (100–200 μM) to visualize diffusion, and the following substances (in mM): 8 KCl, 8 Na- gluconate, 2 $MgCl_2$, 10 HEPES, 4.38 $CaCl_2$, 4 K-ATP, and 0.3 Na-GTP, 10 K-BAPTA (1,2-bis(o-aminophenoxy) ethane-N,N,N',N'-tetraacetic acid) with 96 K-gluconate. Osmolality was adjusted to ~285 mOsm, pH was corrected with KOH to 7.2. We voltage clamped astrocytes at −80 mV and confirmed the cell type by a low input resistance (10–20 MΩ) and dye filling of the endfeet circumventing a large section of the arteriole (Supplementary Fig. 4a, b). Importantly, dye filling of the endfeet with a clamping solution of 100 nM $Ca^{2+}$ did not change arteriole tone per se (see Supplementary Fig. 5). We allowed 15–30 min for any internal solution in the extracellular space from the patching process to washout from the tissue and for the arteriole tone to return to baseline. Then, while in stable whole-cell configuration, we repeated the electric stimulation to compare to the first response.

## Awake in vivo two-photon imaging of acute cranial window
Male mice were used according to the protocol previously developed by our laboratory[11]. Briefly, on day 1, under isoflurane anesthesia (induction 4%, maintenance 1.5–2%), pain control (buprenorphine 0.05 mg/kg) and antibiotic premedication (enrofloxacin 2.5 mg/kg), a light (0.5 g) titanium headbar was glued on the occipital bone under aseptic conditions with a three-component dental glue (C&B Metabond, Parkell Inc, NY, USA) and dental cement (OrthoJet Acrylic Resin, Land Dental MFG. CO., Inc., IL, USA). After a recovery period of 24 h, the mice were trained on days 2 and 3 to habituate to imaging and air puff stimulation to the whiskers while head fixed but free to move on a passive, air supported spherical Styrofoam ball in a light tight environment (Fig. 1a). After 15 min of rest, 5 s continuous air puffs (30 psi, Picospritzer III, General Valve) were delivered to the contralateral whiskers 15 times every 10 s, 15 s, 30 s, and 60 s, respectively on day 2 and 3. On day 4, a craniotomy was performed over the barrel cortex. Bone and dura were gently removed. For *Aldh1l1*-Cre/ERT2 × CAG-LSL-Gq-DREADD mice, Rhod-2/AM (15 μM) in aCSF was incubated on the

brain surface for 45 min. Next, the window was fully sealed with a coverslip. In experiments where astrocyte endfeet were labeled with a red fluorescent dye, 0.2 mL 5% FITC dextran solution was injected into the tail vein immediately before imaging, whereas Rhodamine-B dextran (0.2 mL, 2.3%) was injected when astrocytes or neurons expressed GCaMP6. The animals were transferred to the imaging rig, where imaging started ~30 min after the animals awoke. Behavior was continuously monitored by a camera detecting the light of an infrared LED and was recorded during imaging. A 16× or a 40× water-immersion objective was positioned square to the surface of the window. Imaging was performed at 3.91 or at 0.49 Hz when drug application was recorded. Rhod-2/AM only loaded astrocytes in mice in vivo (Fig. 4a) as neuronal compartments were neither labeled at rest nor appeared in response to stimulation[11].

### Awake in vivo two-photon imaging of chronic cranial window

Male, 3–4 months old c57bl/6 and *Aldh1l1*-Cre/ERT2 × R26-Lck-GCaMP6f mice were used. The surgical procedure was based on Tournissac et al. with modifications[90]. For head bar installation mice were premedicated with dexamethasone (2 h prior, 6 mg/kg s.c.). Under isoflurane anesthesia and aseptic conditions, the parietal, occipital and temporal bones were carefully separated from the surrounding soft tissue and a U-shaped metal headbar with 2 lateral arms was cemented to the occipital and temporal bones by a UV glue (Ivoclar vivident Tetric Evoflow). A conical cement wall was built around the right parietal cortex. The surgery was supported with antibiotic (enrofloxacin), analgesic (buprenorphen 0.1 mg/kg s.c.) and anti-inflammatory (meloxicam 5 mg/kg s.c.) medication. After a minimum of 5-day recovery, mice were premedicated with dexamethasone (2–4 h prior), enrofloxacin, meloxicam, buprenorphine and anesthetized with a mixture of ketamine (100 mg/kg) and medotomidine (0.5 mg/kg) i.p. The mice were temperature controlled and inhaled 100% $O_2$ during cranial window implantation. A 3 mm circular area was removed from the parietal bone over the whisker barrel area, leaving the dura intact. To prevent bone regrowth, a custom-cut T-shaped coverslip (inner diameter: 3 mm; outer diameter: 3.5 mm, central thickness: 250 μm)(Laser Micromachining Ltd.) was sealed into the hole with UV cement. Anesthesia was reversed with atipamezole (0.5 mg/kg s.c.). Post-operative supportive therapy included a heated cage with $O_2$-enriched air (until the mouse turned active) and recovery jello for diet. For CalEx, and mutant CalEx control mice, a tiny slit was cut in the dura, through which the viral construct was injected into the barrel cortex before sealing the craniotomy. Six weeks later, mice were trained the same as in the acute cranial window experiments applying 5 s continuous air puffs to the whiskers. For ball movement and pupil diameter recording 2 near-infrared LEDs (780 nm) and 2 infrared-sensitive cameras (Basler) were pointed at a small area of the Styrofoam ball and the right eye of the mouse, respectively. We recorded pupil + ball movement at 40 Hz, while ball movement for 2P imaging was recorded at 62.5 Hz with the software Pylon Viewer (version 6.2.4.9387) by Basler. An open-source pulse train generator (Sanworks, Pulse Pal v2) was configured to synchronize trigger signals between the cameras, LEDs, the Picospritzer III and the microscope.

### CalEx Plasmid construction

To generate the AAV astrocyte-targeted expression construct of HA-tagged CalEx, the DNA sequence encoding hPMCA2w/b was PCR amplified from pZac2.1-GfaABC1D-mCherry-hPMCA2w/b (a gift from Baljit Khakh (Addgene plasmid # 111568 ; http://n2t.net/addgene: 111568; RRID:Addgene_111568) with primers designed to incorporate a N-terminal HA tag encoding the amino acids YPYDVPDYA and used to replace mCherry-PMCA2w/b in the same backbone at the 5' NheI and 3' XbaI restriction sites using the NEBuilder Hifi DNA assembly kit, generating pZac2.1-GfaABC1D-HA-hPMCA2w/b[23]. For the control construct, the critical $Ca^{2+}$ binding site residue glutamic acid 457 was

converted to Alanine using PCR site-directed mutagenesis, generating a mutant control CalEx (hPMCA2w/b (E457A)) incapable of binding or membrane transport of $Ca^{2+}$[36,37]. The DNA sequences of all constructs were verified by Sanger DNA sequencing.

### AAV production

AAV viral vectors were generated using the methods of Challis et al.[91]. Briefly, 293FT cells (ThermoFisher Scientific) were grown to ~90% confluency in Corning hyperflasks (Corning) and co-transfected with 129 μg pHELPER (Agilent), 238 μg rep-cap plasmid encoding AAV5 capsid proteins (pAAV2/5 was a gift from Melina Fan (Addgene plasmid # 104964 ; http://n2t.net/addgene:104964; RRID:Addgene_104964) and 64.6 μg of either the CalEx plasmid (pZac2.1-GfaABC1D-HA-hPMCA2w/b) or control (pZac2.1-GfaABC1D-HA-hPMCA2w/b (E457A)) using the PEIpro transfection reagent (Polyplus). The AAV viral vector purification, concentration and titer determination was performed as previously described[91–93].

### AAV CalEx injection

A viral cocktail containing AAV2/5.GfaABC$_1$D GCaMP6f ($3 \times 10^{12}$ gc/ml, Addgene) for astrocyte $Ca^{2+}$ detection or AAV9.hSyn.GCaMP6f ($3 \times 10^{12}$ gc/ml, Addgene) for neuronal $Ca^{2+}$ detection was mixed with the astrocyte-targeted CalEx encoding construct AAV2/5.GfaABC1D-HA-hPMCA2w/b ($1.18 \times 10^{13}$ gc/ml) in a 1:3 ratio and delivered by a Nanoject III (Drummond Scientific, Broomall, PA, USA) in a volume of 400 nL. For the best possible control, we designed a mutated version of the CalEx virus in which changing the critical glutamic acid residue in TM4 (E457) to Alanine to render the pump incapable of transporting $Ca^{2+}$. Training and imaging took place 6 weeks later, because this is sufficient time to achieve functionally effective CalEx expression[23].

### Immunohistochemistry

Slices were collected from control and CalEx mice at the end of the in vivo experiments and post-fixed in 4% paraformaldehyde overnight and then cryoprotected in 30% sucrose for another 24 h. Sections were cut on a cryostat to 30 μm and free-floated in PBS. The sections were first washed in PBS (phosphate buffered saline) for 20 min, followed by 3 × 5 min washes in PBS; agitated at room temperature. Sections were permeabilized in 0.1% Triton X in PBS for 10 min. Anti-HA (rabbit) primary antibody (1:100: Abcam) was diluted in 2% BSA in PBS and incubated for 1 h and agitated at room temperature. After primary antibody incubation, sections were washed three times with PBS and then incubated for 1 h with Alexa Fluor 555 (1:1000: Invitrogen) diluted in 2% BSA in PBS and agitated at room temperature. Sections were then washed again three times with PBS before mounting coverslips with ImmunoMount (Epredia) and read on a confocal fluorescence microscope (Leica TCS SP8, Leica Microsystems).

### Functional hyperemia

A two-photon imaging session started by confirming Rhod-2/AM, GCaMP6 or Gq DREADD-linked mCitrine expression and identifying the vascular network by z-stack imaging to a depth of ~300 μm with a 16× objective. Pial arterioles were distinguished from veins by spontaneous vasomotion and by positive responses to contralateral whisker air puff. Penetrating arteriole branches of the pial arteries were tracked and imaged at a depth of 50–250 μm with a 40× or a 16× objective. Air puff was delivered through 2 glass capillary tubes positioned to deflect as many whiskers as possible without blowing air on the face (Fig. 1a). A 10 s baseline period was recorded after which a continuous or an intermittent (4 Hz, 125 ms pulses) 5 s or 30 s air puff (30psi) was employed. Stimulation was repeated 3–6 times and averaged for each arteriole. In case of chronic cranial window experiments, on the day before two-photon imaging we mapped the cortical center of activation to whisker stimulation by collecting intrinsic optical signals (IOS) of hemoglobin under the cranial window. We flashed green

light with a stable light-emitting diode (M530L3, Thorlabs Inc.) equipped with filtering (Semrock) and collimation optics (Thorlabs) at 10 Hz and collected reflected light with a Basler Ace U acA1920-40 µm camera with Fujinon HF50XA-5M lens (pixel size = 5.86 µm), during 30 s whisker stimulations. IOS was recorded with Pylon Viewer (version 5.0.11.10913) by Basler. Penetrating arterioles for two-photon imaging were selected from a region where green light reflectance showed a major decrease during the period of whisker stimulation overlapping with the expression of the viral construct (good neuronal or astrocytic GCaMP6 expression). We measured pupil diameter changes in a separate set of trials with no stimulation, 30 s ipsilateral air puff to the body, or 30 s whisker stimulation while the pupils were constricted to mid-size with a low intensity continuous green light in the background.

**Data processing and statistical analysis**
Ca²⁺ signals were analyzed in ImageJ 2.9.0 (NIH) and calculated in Prism software 9.4.0 (GraphPad Software LLC., La Jolla, CA) as follows: $\Delta F/F = ((F_1 - F_0)/F_0) \times 100$, where F is fluorescence, 1 is at any given time point, and 0 is an average baseline value of 60 s ex vivo or 2 s in vivo pre stimulus. Ca²⁺ responses in brain slices were obtained by selecting a region of interest (ROI) and running the 'intensity vs. time monitor' plugin over the following compartments: (1) neuropil was outlined as an acellular region next to the region of vasodilation on the side closer to the stimulating electrode, (2) neuron soma responses were the average of 5 neuronal cell bodies visible in the imaging plane closest to the arteriole, (3) astrocyte soma signal was the average of 1-3 cells closest to the arteriole, (4) astrocyte endfoot was chosen as a clearly visible continuous structure adjacent to the area of vasodilation. The same ROIs were used after patching in vitro or after drug administration in vivo. Astrocyte GCaMP6 signals of endfeet and processes in all experiments were outlined by including regions around the arteriole where an increase in the fluorescent signal was visible after stimulation onset. On neuronal GCaMP6 recordings ROI was determined as a 5000 µm² polygonal perivascular area avoiding the shadow of the proximal section of the vessel.

In the absence of stimulus-evoked fluorescence changes, areas of basal GCaMP6 fluorescence were selected. Ca²⁺ signals are labeled as $\Delta F/F_c$ in control conditions, $\Delta F/F_{BAPTA}$ after BAPTA loading. Pre-processing of intraluminal dye recordings in ex vivo slices included xy-motion correction, median filter (0.5-2 pixel radius), and Gaussian filter (1-4 pixel radius) performed by ImageJ plugins. Intraluminal area was selected by intensity thresholding and quantified using the 'Analyze particles' function of ImageJ.

To determine the cross-sectional area of intraluminal dye-loaded arterioles in vivo, we used three independent methods to outline the edges of the vascular lumen. For Gq-DREADD (Fig. 4e, f) and MSPPOH experiments (Fig. 6i) we used manual outlining of individual images via Amazon Mechanical Turk (see section below), for thinned skull, and acute window experiments (Fig. 1a-h, Fig. 5a-g, and Supplementary Figs. 1a, 3), we used ImageJ analysis (see above). For CalEx experiments and Aldh1l1-Cre/ERT2 × R26-Lck-GCaMP6f chronic window experiments (Supplementary Fig 2d-f and Fig. 2), for each recording, we quantified cross-sectional area of the imaged penetrating arteriole using the ImageJ analysis approach outlined above and again by using a semi-automated tracking approach detailed in Haidey et al.[82] (see below), and then averaged the two. Briefly, the latter approach involved: (1) defining regions of interest from blood-vessel cross-sections at different imaging angles to create tracking templates; (2) selecting a bounding box constraining the tracking algorithm to the maximum observed dilation in the recording; (3) applying MATLAB's implementation of the Thirion's DEMONS algorithm (with default options) for non-rigid image matching to estimate ROI deformation within the bounding box relative to the tracking template, which provides the cross-sectional area estimate. A color-coded image of the input template with the detected edges is shown in Supplementary

Fig. 12i. Arteriole diameter changes were quantified as the relative change in the area of intraluminal dye along the longitudinal section (in slice) or calculated from the cross-sectional area (in vivo) of the vascular lumen, assuming a circular shape, as follows: diameter = 2× √(area/π). Stimulation-evoked vasodilation was first calculated as Δ diameter ($\Delta d$) = (diameter₁ − diameter₀)/diameter₀) × 100 (%), where diameter₁ is the diameter of the vascular lumen at any time point and diameter₀ is the average baseline vascular lumen diameter 2 s prior to stimulation onset. Response latency was determined as the time from stimulation onset until 2 consecutive values exceeded 3 standard deviations of the 2 s baseline. All peak responses during topical C21 application (first 10 min from Ca²⁺ increase onset) were compared to the peak value of the 500 s baseline period (see Fig. 4c). AUC values were calculated from Δd traces as the summary of all positive peaks over the baseline value for 35 min from stimulation onset. For data presentation and statistical analysis averaged arteriole diameter traces of whisker stimulation trials served as single data points as we expected a uniform and reproducible response across trials. For astrocyte and neuronal Ca²⁺ response analysis we used data points of individual trails because we considered the response of an individual cell less stereotypical and predictable (based on citation[11]).

Event-based analyses of Ca²⁺ signals in neuronal compartments were performed by an automated analysis toolkit (https://github.com/yu-lab-vt/AQuA)[39] on 5000 µm² peri-arteriole ROIs of image stacks recorded at 7.82 Hz. A Gaussian spatial filter was applied blindly to each trial (1-4 standard deviation of the mean) to optimize the fidelity of event detection. Movement-related event detection artefacts were excluded. Absolute event frequency (calculated as the average number of events for each sec), peak fluorescence value (dF/F Max), integrated Ca²⁺ event value (AUC dF/F), event area size (µm²) and the duration (time of rise and fall between the 10% of peak) were summarized from the analysis output.

Locomotion analysis: Recordings of a small area of the Styrofoam ball were analyzed by a custom MATLAB script to quantify ball movement on a relative locomotion scale where '0' is assigned to the least movement (stationary ball) and 1 is the highest speed an animal reached during the experiment.

To quantify locomotion activity from video recordings of a small region of the Styrofoam ball, we implemented a simple algorithm in MATLAB[94]: briefly, the image stack was read in and duplicated; the first frame from 'Stack1' and the last frame from 'Stack2' were deleted to yield an image stack that reported pixel motion within the recorded video (which directly related to the animal's location); a pixel-by-pixel subtraction was performed for every image in the stack (i.e., Stack1−Stack2); mean gray value was then plotted against time for the resulting stack, providing an estimate of relative locomotion within the imaging trial.

Pupil diameter analysis: Pupil diameter changes were recorded at 40 Hz using an infrared camera mounted close to the animal's eye. To quantify pupil diameter from the videos, we used single-animal Dee-pLabCut to train a model to track four points on the circumference of the pupil (two along the vertical axis and two along the horizontal axis)[95]. The following parameters were used: TrainingFraction = 0.95; net_type = resnet_50; augmenter_type = imgaug; max_iters = 500,000. Tracking performance was assessed by visual examination of the processed videos and the filtered position traces; frames where the animal blinked (<0.5%) were discarded. The average of the horizontal and vertical distance was used as pupil diameter. A custom MATLAB script (version 2021b) was used to determine pairwise distances and perform median filtering of the diameter traces (https://github.com/tpgovind/Locomotion-Pupil-Tracking, https://doi.org/10.5281/zenodo.7246747).

Refer to Supplementary Tables 1−6 for the statistical details that pertain to the six main text figures. Statistical analyses were performed on averaged trials of PA responses and on Ca²⁺ responses of individual

stimulation trials in vivo. Normality was tested with Shapiro–Wilk test. Normally distributed data were evaluated with a paired or unpaired Student's t-test, and non-normally distributed with Mann–Whitney (Fig. 3a, b) or Wilcoxon test (astrocyte Ca²⁺ signals in Supplementary Fig. 5). The non-parametric Friedman test and Dunn's multiple comparison test was applied to compare in vivo peak diameter changes on stimulation length (1–5–30 s), astrocyte process Ca²⁺ peak at various time periods of stimulation (Supplementary Fig. 2f) and to compare methods of arteriole cross sectional area analysis (Supplementary Fig. 12b). Mixed effects model with Holm–Sidak's multiple comparison test was used for response latency comparison (Fig. 1e). Response latency of arteriole, endfoot and process Ca²⁺ was analyzed with Kruskal-Wallis test and Dunn's multiple comparison test (Supplementary Fig. 2d, e). One-way ANOVA with Tukey's post hoc test was applied on pupil response comparisons (Fig. 3g) Two-way ANOVA was run to compare the effects of stimulation length (5 s vs 30 s), external manipulations (CalEx, or drug treatments) and their interaction, followed by Tukey's post hoc test for pairwise comparison. Baseline arteriole diameter (intravascular area) changes were analyzed with the two-tailed non-parametric Wilcoxon test. Data are expressed as mean ± standard error of the mean (SEM). We analyzed the effect of locomotion as a co-variate for each trial on group differences in arteriole diameter, astrocyte and neuronal Ca²⁺ signals of the CalEx experiment using generalized linear models in SPSS (IBM SPSS Statistics, Version 25). Every other statistical analysis was performed in Prism 9.4.0 (GraphPad Software, Inc., La Jolla, CA, USA). Levels of significance used were: $*p < 0.05$, $**p < 0.01$, $***p < 0.001$, $****p < 0.0001$.

**Amazon mechanical Turk analysis**

Arteriole diameter changes of the CalEx experiments were analyzed by manual outlining of the FITC dextran loaded vascular lumen frame by frame in an outsourced and randomized fashion, using Amazon Mechanical Turk (Supplementary Fig. 12)[96]. It is an online crowdsourcing website, through which remotely located "crowd-workers" are hired to perform a discrete on-demand task. A custom-made GUI (https://github.com/leomol/vessel-annotation, https://doi.org/10.5281/zenodo.7231728) facilitated the manual outlining of the vascular cross-section over the course of the recording (https://codepen.io/leonardomt/full/jOEvPvY). An annotation task consisted of adjusting a polygonal shape with a fixed number of vertices on the top of the vessel. To keep annotations consistent, crowd-workers could inspect the time evolution of the vessel and their annotations by freely scrolling through the time-lapse images. Annotation tasks were randomly assigned to 10 different crowd-workers, blind to the experiment. Each stack of images was then visually examined and accepted or rejected based on whether the instructions were followed or not, i.e., if outlines were not consistently labeled over time or if there was a clear mismatch in the outlines with respect to the brain vessel. The area of each polygonal shape was then calculated and normalized (Supplementary Fig. 12).

**Reporting summary**

Further information on research design is available in the Nature Portfolio Reporting Summary linked to this article.

## Data availability

Source data are provided with this paper. Raw image files are stored on servers at Hotchkiss Brain Institute owing to their large size. These raw images can be provided from the corresponding author upon request.

## Code availability

The code for the custom-made automated luminal area tracking algorithm is available at: https://senselab.med.yale.edu/modeldb/enterCode?model=266929. The code for the custom-made GUI for the Amazon Mechanical Turk analysis has been deposited to

Zenodo: https://zenodo.org/record/7231728. The code for the pupil tracking and locomotion analysis have been posted to Github and Zenodo: https://github.com/tpgovind/Locomotion-Pupil-Tracking. https://doi.org/10.5281/zenodo.7279708, https://doi.org/10.5281/zenodo.7246747. The code for event-based analyses of Ca²⁺ signals in neuronal compartments: https://github.com/yu-lab-vt/AQuA.

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

## Acknowledgements

Our work is supported by grants to G.R.G from the Canadian Institutes of Health Research (PJT-173468 and FDN-148471) and by the Hotchkiss Brain Institute and the University of Calgary. We appreciate Dr. Marine Tournissac's assistance in establishing the chronic cranial window model. We thank Dr. David Rosenegger for preliminary patch experiments. We thank Drs. Ciaran Murphy-Royal, Brian MacVicar and Rebecca Williams and Darren Clark for their valuable scientific comments. We thank Dr. Vincent Ebacher and the HBI advanced microscopy platform for the confocal microscope images. We thank the CSM Optogenetics Facility for hosting the web services for crowdsourced annotations. We are grateful for Dr. Frank Visser for transgenic mouse sample genotyping. We thank the developers and distributors of ScanImage open-source control and acquisition software for two-photon laser-scanning microscopy.

## Author contributions

A.I. designed and performed experiments, analyzed results, and wrote the paper. M.V. and G.P. performed experiments and analyzed results, C.C. analyzed results, C.H.T. performed experiments, X.Y., and B.S.K., designed CalEx virus construct, F.V. designed the mutant CalEx virus construct, C.B. did immunohistochemistry, L.M. designed platform for crowdsourcing annotations with Amazon Mechanical Turk and supervised analysis. M.N. provided trainee supervision, experiment design, and edited the manuscript. R.J.T. provided trainee supervision, experiment design, and edited the manuscript. G.R.G. conceived the project, designed experiments, and wrote the paper.

## Competing interests

The authors declare no competing interests.
