## [Peer Review File · Nature Communications]

Astrocytes amplify neurovascular coupling to sustained activation of neocortex in awake miceREVIEWER COMMENTS

Reviewer #1 (Remarks to the Author):

This new paper by Grant Gordon's group describes the results of an impressive series of experimental studies aimed to investigate the functional significance of astroglial Ca²⁺ signals induced by increases in local neuronal activity. In particular, the authors re-evaluated the role of astroglial Ca²⁺ elevations in mediating activity-dependent dilations of cortical penetrating arterioles (functional hyperemia). The data presented indicate that the cerebrovascular responses to neuronal activation in the barrel cortex consist of two components – immediate, developing in response to stimuli of short duration; and sustained, recorded in response to longer sensory stimulation. The data are beautifully presented and suggest that the immediate neurovascular response precedes and appears to be independent of astroglial Ca²⁺. In contrast, the results suggest that astrocytic Ca²⁺ signalling contributes to the late component of functional hyperemia, when the neuronal activation is prolonged. The studies are expertly performed, and the data analysis is appropriate, yet, in my opinion, additional control experiments are required to support the conclusions reached by the authors.

1. The major limitation of this study is the lack of neuronal activity data. Although, the authors discuss this limitation, recordings of the cortical neuronal responses to sensory stimulation in conditions when astrocyte Ca²⁺ signalling is manipulated is required to support the conclusions reached. CalEx is a wonderful tool, but expression of high-affinity Ca²⁺ pump in astrocytes may have a significant impact on the ability of neighbouring neurons to respond and maintain increased activity in response to prolonged afferent stimulation. The authors partially address this issue by the experiments in acute brain slices with clamping of astroglial Ca²⁺ and the recordings of vascular responses and intracellular Ca²⁺ in neurons. Yet, the data obtained in these in vitro experiments are not sufficient to rule out the possibility that CalEx expression in astrocytes changes neuronal excitability.
2. The same limitation applies to the experiments with chemogenetic activation of astrocytes using DREADDGq expression and application of C21. It is plausible that activation of astrocytes leads to the release of gliotransmitters which increase the excitability of the neuronal network, augment the neuronal responses, and, therefore, result in larger and longer arteriole dilations.
3. The same limitation also applies to the experiments with NMDAR inhibition and EET production blockade. These experiments should be supplemented by the recordings of the neuronal activity changes in response to sensory stimulation to make sure these are similar in different experimental conditions.

Minor:

1. Title: I am not sure that the term "unleashes" is appropriate in this context. Google gives the following two definitions: "release (a dog) from a leash", and "cause (a strong or violent force) to be released or become unrestrained". I would suggest changing "unleashes" to "triggers", or "is associated with", or similar.
2. Data Presented in Fig 1. Profiles of Ca²⁺ responses recorded using Rhod-2 and GCaMP6s are very different. Rhod-2 records Ca²⁺ elevations that are largely sustained during the 30-s period of stimulation and return back to baseline after the period of stimulation (panel C). GCaMP6s-reported signals return to the baseline before the termination of the stimulus (panel D, and other figures). Perhaps these differences need to be discussed.
3. Lines 339-340: "...and they argue against a NMDAR-nitric oxide link as a rapid mediator in CBF control in awake mice". This conclusion however goes against the bulk of data suggesting that NO produced by neuronal NOS plays a key role in the neurovascular coupling response (PMID 30481531). If the authors wish to defend this point, then they perhaps need to suggest another plausible mechanism of how NO production by cortical interneurons is facilitated in response to

sensory stimulation.

Reviewer #2 (Remarks to the Author):

The paper "Sustained activation of awake mouse neocortex unleashes an astrocyte-mediated amplification of neurovascular coupling" applies a variety of challenging approaches to address the important and unresolved question of astrocytic involvement in neurovascular coupling. Over many years this cell type has been a common suspect in this regulation, due to its morphology that links synapses to vascular surfaces. A major issue is the unpredictability of astrocytic Ca²⁺ changes to stimulation, especially with regards to timing, but also with regards to size and spatial distribution. This paper focus on the Ca²⁺ activity in the end-feet surrounding the penetrating arteriole (PA). The authors of this paper have in several other good publications presented results that showed the unpredictability and lateness of this signal in comparison to PA dilation events. The important and well supported finding presented here is that this Ca²⁺ signal is crucial in prolonging the PA dilatation observed during long-lasting whisker stimulations. An extended dilation of the PA signifies a substantial increase in blood delivered to the tissue supported by this point of entry to the cortex. That astrocytes regulate blood flow at the PA level in this way is novel and moves the field forward from the understanding gained by two independent works that in 2016 argued that this was not the case^{1,2}. I thus find that this is a paper especially relevant to those working in the blood flow regulation field, but also very relevant to all with an interest in human brain mapping or astrocytic contribution to CNS pathologies.

The paper is in general well written and results clearly presented in the figures, though with a few flaws addressed as minor comments. The introduction and discussion refers to relevant papers and relate the findings in a proper way. With regards to the conclusions, they are in general sound though the authors might benefit from a more cautious evaluation of their methods in the discussion section. The genetic approaches to modification of cellular Ca²⁺ levels are very relevant and elegantly used but must be taken cautiously due to the changes they may cause in the complex and diverse regulation of intracellular Ca²⁺ levels.

Major comments:

In the CalEx results the significant reduction in Ca²⁺ levels seem to be based on changes in the responses to 5sec. rather than the 30sec. stimulations. This may be due to a very large variance not evident from the way the average traces are represented in F and G, using SEM around the mean. Still, it is surprising, given the clear reduction in the average trace, that this cannot be reproduced in summarized values. Especially the AUC evaluations where the effect of CalEx expression on 30s stimulations is quite small and not statistically significant. Is it possible that the CalEx effect is primarily to delay the rise time in the astrocyte Ca²⁺ levels? And wouldn't the more precise description of the effect be that the Ca²⁺ rise, diminished/delayed by the CalEx, must be a recruiting factor of the second part of the biphasic dilation? This goes well in hand with the data presented in figure 3 and 4 as well.

With regards to the variance in the astrocytic Ca²⁺ signals several of the authors contributed to a study of the heterogeneity of the Ca²⁺ responses, showing that roughly 50% of the astrocyte endfeet surrounding PA responded to a whisker stimulation³. How is that reflected in the averaged traces? And in the max peak and AUC Ca²⁺ data? Have the authors excluded Zero-responses events or included them in the averages? Were the occurrence of astrocytic PA endfeet Ca²⁺ responses evaluated, and did it change with the three tools applied to modify it (CalEx, Dread, AP5)?

The authors use an air puff mediated whisker stimulation to initiate neurovascular response in the barrel cortex. Though it is mentioned in the methods section that it is given without blowing air in the face of the mouse, the reader would benefit from a brief assurance of the non-startling nature of this

stimulation paradigm from the very beginning of the results section. This is important to underline given that air-puffs are used as startling stimulations which will induce a large Ca²⁺ response in astrocytes^{4,5}. There is an indirect affirmation of potentially startling effect of air puffs in the use of a “untrained body air puff” to startle the animals in supplemental figure 2. The paper would benefit from clearly addressing this issue, preferentially through a small investigation into the connection between these puff stimulations and pupil size or similar⁶.

Just as a stimulation paradigm may influence the vascular and cellular responses through neuromodulators, locomotion will also influence the activity of the barrel cortex⁷. The authors do not address this issue. If responses during locomotion is disregarded this should be clearly stated in the text. If not, it may have confounded the results and added the variance and timing of the vascular and astrocytic Ca²⁺ responses.

In Fig.4 the authors have chosen to present the effect of AP5 on vascular diameter in another set of animals than those used to evaluate the effect on astrocytic Ca²⁺. While this is not disqualifying it is puzzling, given that it appears as if the Ick-GCaM6f measurements were done in combination with Rhodamine D as a vascular marker. Why are the diameter changes not evaluated in the same mice as the Ca²⁺ changes in astrocytes following the AP5 based modification of the NMDAR activity? Finally, I am unconvinced that these experiments show that the effect is through astrocytic NMDAR alone, but the authors also acknowledge this in the discussion.

Minor issues:

Some of the data shown in Fig 1 seems unnecessary to convey the information. Why do we need section C to show the nature of the astrocytic Ca²⁺ when it is similar to and as clearly visible from the data presented in panel D? I am not certain the interest if the readers stretch to the sensitivity of Rhod2AM compared to the GdCaMP6s as an indicator on astrocytic Ca²⁺ levels.

There seems to be a redundancy in the text referring the results from Fig2E. All text from line 169 “.Again, peak...” to “stimulation (Fig. 2E).” in line 173 refer to what is already clear presented from line 164.

In a few places the large multi-figures could benefit from a more systematic organization. Examples: Fig.1 B “arteriole diameter” is written above the pictures in, but in C “astrocyte endfoot Ca²⁺” is written below the pictures in. In Fig.2 E F and G aligning the text headlines would give a less “jumpy” impression. Fig.4 F, two air puff capillaries in cartoon.

I find it peculiar that the author refers to the studies that show biphasic response curves as recent (line 64), since they are from 2011 and 2013 respectively.

On the figure it is written Rhodamine D as an abbreviation for Rhodamine-B-dextran. I find that to be suboptimal solution. Maybe Rhodamine B or Rhodamine-dex?.

Supplemental fig4 A is very difficult to read, especially when printed out on A4 format.

Reference List

- 1 Mishra, A. et al. Astrocytes mediate neurovascular signaling to capillary pericytes but not to arterioles. *Nat Neurosci* 19, 1619-1627, doi:10.1038/nn.4428 (2016).
- 2 Biesecker, K. R. et al. Glial Cell Calcium Signaling Mediates Capillary Regulation of Blood Flow in the Retina. *J Neurosci* 36, 9435-9445, doi:10.1523/JNEUROSCI.1782-16.2016 (2016).
- 3 Sharma, K., Gordon, G. R. J. & Tran, C. H. T. Heterogeneity of Sensory-Induced Astrocytic Ca⁽²⁺⁾ Dynamics During Functional Hyperemia. *Frontiers in physiology* 11, 611884, doi:10.3389/fphys.2020.611884 (2020).
- 4 Ding, F. et al. alpha1-Adrenergic receptors mediate coordinated Ca²⁺ signaling of cortical

astrocytes in awake, behaving mice. *Cell Calcium* 54, 387-394, doi:10.1016/j.ceca.2013.09.001 (2013).

5 Zuend, M. et al. Arousal-induced cortical activity triggers lactate release from astrocytes. *Nat Metab* 2, 179-191, doi:10.1038/s42255-020-0170-4 (2020).

6 Reimer, J. et al. Pupil fluctuations track rapid changes in adrenergic and cholinergic activity in cortex. *Nature communications* 7, 13289, doi:10.1038/ncomms13289 (2016).

7 Shimaoka, D., Harris, K. D. & Carandini, M. Effects of Arousal on Mouse Sensory Cortex Depend on Modality. *Cell reports* 25, 3230, doi:10.1016/j.celrep.2018.11.105 (2018).

Reviewer #3 (Remarks to the Author):

The manuscript by Institoris et al tackles an important question in the field of neurovascular coupling: what is the role of astrocytes in triggering or modulating vascular responses? Most studies have used correlations between blood flow changes and astrocyte signals to infer or discard any role for these cells. Initially, mostly somatic signals were measured for technical reasons, resulting in misleading interpretations. Then, monitoring of astrocyte processes revealed that astrocytes could potentially be involved in NVC whereas the persistence of functional hyperemia in IP3R2 KO animals questioned this hypothesis. Here, the authors used an elegant approach to silence astrocyte calcium and observed that it modulates NVC. I find the topic interesting, but several issues need to be addressed. Major Comments:

1) In their introduction, the authors overstate that anesthetics alter astrocyte responses. The problem is widely known but more subtle, as it depends on the drug and concentration used. Data from anesthetized animals are certainly as relevant as those from the current study slice experiments. I therefore suggest that the authors introduce the topic with the paper by Schultz et al., which was the first to seriously correlate the activation of astrocytes with the stimulation intensity and the appearance of a delayed vascular response. The spatial distribution of the delayed response was also investigated by Chen et al. 2014. Finally, in the current study, all the experiments in awake mice are performed immediately after surgery, starting 30 min after the animals awoke. It is certain that recovery from surgery per se requires more than 30 mins as well as the complete recovery from isoflurane anesthesia (Shirey et al. 2015). The authors should therefore consider that their preparation is not fully normal, but this does not lower the value of their findings.

2) The authors mostly show average trace data from many animals. It emphasizes the reproducibility between animals but masks resting state fluctuations and the precise determination of onsets. For example, in Fig1C Ca²⁺ responses to 30 s stimulations seem to start almost immediately whereas in Fig1D a delay is observed, and the response lags the beginning of the astrocyte process signal. The authors quantify the latency as "the time from stimulation onset until 2 consecutive values exceed 3SD of the 2s baseline". It is probably too cautious, and I suspect that Ca²⁺ from processes would precede endfoot Ca²⁺ if the onset were determined with 2SD of the 2s baseline or using other approaches (the observation also applies to Fig4). Could the author check the hypothesis? Note that the conclusion of this result section (and Fig1) could be that the mechanisms underlying both early and late vascular responses start at the level of the SMCs and not astrocytes.

3) CalEx experiments:

-To clamp astrocyte Ca²⁺, the authors use AAVs. Suppl Fig2 shows that response latencies in both control and CalEx animals seem larger than in Rhod2 or Aldh1GCaMP6 animals. Could the authors quantify and/or discuss these differences?

-Fig2E shows that the effect of CalEx on average traces of the initial dilation peak differ for 5s and 30s stimulations. Why is this difference not observed in the plots (peak $\Delta D/D$)? There should be a better match between the plots and the traces shown, in particular as the authors conclude that CalEx acts primarily on the late component of functional hyperemia.

4) In vitro experiments:

Could the authors clarify what stimulation protocols are used to trigger dilation in vitro. In Suppl Fig 4 it seems that different stimulation frequencies are used for 30s (Suppl Fig4, high frequency) and 5s stimulations (Suppl Fig5, theta burst). Given this assumption is right, why theta burst stimulation was

not systematically? It would ease the comparison.

5) Chemogenetic activation:

-The effect of compound 21 on the late phase dilation is nice. As the effect on resting calcium recovers during the 1hr pretreatment, could the authors show if the dilation increase is matched by a Ca^{2+} response increase in astrocytes.

6) NMDA receptors:

1 mM DLAP5 blocks Ca^{2+} responses in astrocytes and the delayed dilation. What is clearly missing is the effect on neurons as prolonged stimulation is likely to progressively activate neuronal NMDA channels. Could the authors control the extent to which the neuronal response is blocked by DL-AP5?

We thank the reviews for their thorough and constructive assessment of our manuscript. We have taken extra time in our revision to not only address the previous questions but improve the previous data by repeating some of the key experiments in a better model with better controls, and with better behavioural tracking. To summarize for all reviewers:

- 1) We have completely reimagined the critical astrocyte CalEx experiment for a number of reasons. Previously, this experiment was done in an acute cranial window model, with potential side-effects from the surgery. We have now redone the experiment in a chronic, fully sealed cranial window with 4-6 weeks of recovery before imaging. In this new preparation, the vessel and calcium responses are larger, more consistent and the results are crystal clear: astrocyte CalEx reduces only the late phase of sustained functional hyperemia. Additionally, we needed to make more CalEx AAV and when doing so, we improved on the tool in two important ways. First, the previous CalEx was tagged with mCherry, which is difficult to detect on the 2P. Furthermore, the CalEx control virus lacked a proper protein control and simply packaged tdTomato. When trying to image GCaMP6 combined with tdTomato in the same cell, we have noticed that the GCaMP signal is much weaker than compared to GCaMP expression alone. We suspect that the mCherry or tdTomato absorbed a significant fraction of the GCaMP emission, decreasing signal-to-noise and making it difficult to detect the effect of CalEx on evoked astrocyte calcium signals to whisker stim. Thus, we eliminated the red fluorescent protein from our new viruses and instead inserted an HA tag for post-hoc immunofluorescence detection to confirm the expression of CalEx or its control in each animal. In our new data set, the control calcium signals are much larger and clearer, and the effect of CalEx is robust during whisker stim. Second, we improved the CalEx control virus. We discovered in the literature that the high affinity Plasma Membrane Calcium ATPase that constitutes CalEx (hPMCA2w/b) can be silenced with a single amino acid substitution (E457A) rendering the transporter ineffective. We made this mutation and have employed this CalEx mutant control in our experiments, which is no longer a simple empty vector control. We find that the mutant CalEx control has little impact on evoked astrocyte calcium signals to whisker stimulation or startle whereas CalEx has a profound effect. This new data replaces the old CalEx experiment.
- 2) Mainly in response to R1, we performed several sets of new experiments tracking neuronal activity via AAV.hSyn.GCaMP6f in the following conditions: 1) Astrocyte CalEx, Astrocyte Gq-DREADD and AP5. This new data has been added to the paper. The details are discussed in the specific comments below but the take-home message is that any changes in neural activity caused by our astrocyte manipulations (CalEx and Gq-DREADD) does not adequately explain the preferential effects on the late component of sustained functional hyperemia we observed.
- 3) Mainly in response to R2, we have improved our behavioural tracking of the animals during functional hyperemia trials and now include a thorough account of both locomotion and pupil diameter in the new CalEx and CalEx mutant control experiments. In brief, pupil diameter significantly increases when we startle the animal, and pupil dilation increases less so in response to whisker stimulation. For locomotion, overall running is not different between the two groups; thus, locomotion is unlikely to explain the difference in late-phase vasodilation between CalEx and control.

Reviewer #1 (Remarks to the Author):

This new paper by Grant Gordon's group describes the results of an impressive series of experimental studies aimed to investigate the functional significance of astroglial Ca²⁺ signals induced by increases in local neuronal activity. In particular, the authors re-evaluated the role of astroglial Ca²⁺ elevations in mediating activity-dependent dilations of cortical penetrating arterioles (functional hyperemia). The data presented indicate that the cerebrovascular responses to neuronal activation in the barrel cortex consist of two components – immediate, developing in response to stimuli of short duration; and sustained, recorded in response to longer sensory stimulation. The data are beautifully presented and suggest that the immediate neurovascular response precedes and appears to be independent of astroglial Ca²⁺. In contrast, the results suggest that astrocytic Ca²⁺ signalling contributes to the late component of functional hyperemia, when the neuronal activation is prolonged. The studies are expertly performed, and the data analysis is appropriate, yet, in my opinion, additional control experiments are required to support the conclusions reached by the authors.

1. The major limitation of this study is the lack of neuronal activity data. Although, the authors discuss this limitation, recordings of the cortical neuronal responses to sensory stimulation in conditions when astrocyte Ca²⁺ signalling is manipulated is required to support the conclusions reached. CalEx is a wonderful tool, but expression of high-affinity Ca²⁺ pump in astrocytes may have a significant impact on the ability of neighbouring neurons to respond and maintain increased activity in response to prolonged afferent stimulation. The authors partially address this issue by the experiments in acute brain slices with clamping of astroglial Ca²⁺ and the recordings of vascular responses and intracellular Ca²⁺ in neurons. Yet, the data obtained in these in vitro experiments are not sufficient to rule out the possibility that CalEx expression in astrocytes changes neuronal excitability.

We thank the review for this important comment. We expect that CalEx will affect some aspect of neuronal function because attenuating astrocyte calcium signaling generally with this approach or with intracellular BAPTA is known to alter neuronal properties. Indeed, there are now 10 years of very good studies on this point, including by Yu and Khakh who made CalEx. Thus, the most critical aspect to determine is whether any changes in neural activity detected by astrocyte CalEx can reasonably account for the preferential effect on the late component of functional hyperemia. We conducted a new set of experiments in a chronic cranial window prep with astrocytes expressing CalEx or our new CalEx mutant control via AAV (see above), along with neuronal (hSyn) GCaMP6f AAV, and performed 5sec and 30sec whisker stimulations. We analyzed this data in two different ways. First, we measured the bulk neural calcium signals from all neuronal compartments non-specifically immediately around the arteriole in a (5000µm² area). Importantly, we found no difference in this type of evoked neuronal calcium between astrocyte CalEx or control (see Fig 2o-p). For a more sophisticated approach, we additionally quantified all neuronal calcium events in the imaging field of view using AQuA, which is an event-based algorithm that finds regions of calcium elevation in time and space. While this tool was designed for astrocyte calcium transients, we found it effective for detecting and measuring neuronal calcium transients too. This technique was able to detect significant differences in both baseline and evoked neural calcium signals. In sum, during whisker stimulation astrocyte CalEx appears to decrease the peak (dFF Max) and slightly increase the area-under-the-curve, of neuronal calcium transients (see Suppl. Fig. 7). Importantly though, there is no preferential effect from any of these neuronal calcium measurements

on the late versus early component of functional hyperemia. Thus, we do not believe these changes adequately explain why astrocyte CalEx only affects the late phase of functional hyperemia. As it is the late phase when astrocyte calcium is predominately elevated in control conditions, the most parsimonious explanation is that astrocyte CalEx is affecting direct communication from the astrocyte to the arteriole. However, we cannot know this for sure and we have this point in the discussion, similar to our previous submission (P24, Lines: 436-440):

“However, it was expected that astrocyte CalEx would have some impact on the local neural activity given the literature and that CalEx targets to both perisynaptic processes as well as endfeet. Indeed, our detailed analysis of neuronal Ca²⁺ transients in the parenchyma showed some effects, yet without a preference for the late phase of 30sec stimulation. In either scenario (direct effect on the vessel or indirect effect via neural activity), our data points to an important role for astrocyte Ca²⁺ in this amplification phenomenon.”

2. The same limitation applies to the experiments with chemogenetic activation of astrocytes using DREADDGq expression and application of C21. It is plausible that activation of astrocytes leads to the release of gliotransmitters which increase the excitability of the neuronal network, augment the neuronal responses, and, therefore, result in larger and longer arteriole dilations.

Similar to CalEx above, we conducted a new set of experiments in astrocyte Gq-DREADD transgenic mice, along with neuronal (hSyn) GCaMP6f AAV, and performed 5sec and 30sec whisker stimulations. We analyzed the bulk neural calcium signals arising from various neuronal compartments immediately around the arteriole in a 5000 μm^2 area and found no significant difference between neuronal calcium responses measured in the presence of C21 vs before (Fig. 4g-i). We refrained from performing AQUA on this data set as it is very time consuming. We feel the caveat of potential neural influences on the vessel is best dealt with by the CalEx experiment because that is the astrocyte “necessity” test. Multiple studies have shown the “sufficiency” of astrocytes to change arteriole diameter *in vivo* (Wang et al., 2006 Nat Neuro, Masamoto 2015 Sci Rep, Hatakeyama et al., 2021 JCBFM). It was the necessity test that was lacking which our study now addresses.

3. The same limitation also applies to the experiments with NMDAR inhibition and EET production blockade. These experiments should be supplemented by the recordings of the neuronal activity changes in response to sensory stimulation to make sure these are similar in different experimental conditions.

For the topical application of the NMDA receptor antagonist AP5, in which we observed a complete elimination of the late phase of functional hyperemia, we agree with the reviewer that more experiments are necessary to better understand what is happening to neural activity. Specifically, it is important to know if AP5 abolishes the late phase of neural activity during sustained sensory stimulation, because if true, then astrocytes need not be recruited in late functional hyperemia in a NMDA receptor-dependent manner. Thus, we performed additional *in vivo* experiments where we measured knock-in Thy1 GCaMP6f mice, and tested 5sec and 30sec whisker stimulations in the presence or absence of AP5. When measuring the bulk neuronal calcium signal directly around the arteriole (5000 micron²), we found that AP5 reduced the peak of neuronal calcium signal to both 5sec and 30sec

whisker stimulation. Notably, the effect of AP5 on the 30sec response was uniform across the entire length of the signal, again showing no preferential effect on the late phase. This data is added to the results on page 20-21 lines 358-369 and appears in Fig 5h. Nevertheless, as this is a non-specific calcium signal, we cannot exclude the possibility that at least part of the reduction in late phase functional hyperemia is attributed to reduced neural activity. This may explain the more pronounced reduction in the late phase of hyperemia by AP5 compared to the effect of CalEx. We have stated these points in the discussion on page 24 line 454-457.

However, we feel that testing for changes in neural activity in the EETs synthesis blocker MSPPOH is not as critical. EETs are well described vascular mediators acting directly on microvasculature to control relaxation of vascular smooth muscle (Munzenmaier and Harder, 2000, Potente et al., 2003, Pozzi et al., 2005, Zhang and Harder, 2002). Any effects of MSPPOH on neural activity is expected to be relatively minor and uninformative.

Minor:

1. Title: I am not sure that the term "unleashes" is appropriate in this context. Google gives the following two definitions: "release (a dog) from a leash", and "cause (a strong or violent force) to be released or become unrestrained". I would suggest changing "unleashes" to "triggers", or "is associated with", or similar.

We thank the reviewer for this comment. We have changed the title to:

Astrocytes amplify neurovascular coupling to sustained activation of neocortex in awake mice

2. Data Presented in Fig 1. Profiles of Ca²⁺ responses recorded using Rhod-2 and GCaMP6s are very different. Rhod-2 records Ca²⁺ elevations that are largely sustained during the 30-s period of stimulation and return back to baseline after the period of stimulation (panel C). GCaMP6s-reported signals return to the baseline before the termination of the stimulus (panel D, and other figures). Perhaps these differences need to be discussed.

The reviewer makes an astute observation that we also noticed. Rhod-2 is well described to load mitochondria in addition to the cytosol, whereas GCaMP is only localized to the cytosol. This is one potential reason why the Rhod-2 signal lasts longer. However, an involvement of astrocyte mitochondria in the effects we describe is not pursued in the current submission. With all our new data including a chronic window with astrocyte Ick-GCaMP6f, we feel the Rhod-2 data do not add much to the paper and we have decided to remove it. This was also suggested by R3.

3.Lines 339-340: "...and they argue against a NMDAR-nitric oxide link as a rapid mediator in CBF control in awake mice". This conclusion however goes against the bulk of data suggesting that NO produced by neuronal NOS plays a key role in the neurovascular coupling response (PMID 30481531). If the authors wish to defend this point, then they perhaps need to suggest another plausible mechanism of how NO production by cortical interneurons is facilitated in response to sensory stimulation.

We agree with the reviewer that there are a lot of fundamental papers that state NMDA receptor activation drives NO production from nNOS as an early/rapid mediator of functional hyperemia. While there is paucity of work exploring this pathway in awake and active animals, and the work that does exist in anesthetized animals rarely reports the kinetics of the CBF/arteriole response, we agree with the reviewer that it is best not to make this statement as we do not have a good alternative. We therefore removed the statement.

Reviewer #2 (Remarks to the Author):

The paper “Sustained activation of awake mouse neocortex unleashes an astrocyte-mediated amplification of neurovascular coupling” applies a variety of challenging approaches to address the important and unresolved question of astrocytic involvement in neurovascular coupling. Over many years this cell type has been a common suspect in this regulation, due to its morphology that links synapses to vascular surfaces. A major issue is the unpredictability of astrocytic Ca²⁺ changes to stimulation, especially with regards to timing, but also with regards to size and spatial distribution. This paper focus on the Ca²⁺ activity in the end-feet surrounding the penetrating arteriole (PA). The authors of this paper have in several other good publications presented results that showed the unpredictability and lateness of this signal in comparison to PA dilation events. The important and well supported finding presented here is that this Ca²⁺ signal is crucial in prolonging the PA dilatation observed during long-lasting whisker stimulations. An extended dilation of the PA signifies a substantial increase in blood delivered to the tissue supported by this point of entry to the cortex. That astrocytes regulate blood flow at the PA level in this way is novel and moves the field forward from the understanding gained by two independent works that in 2016 argued that this was not the case^{1,2}. I thus find that this is a paper especially relevant to those working in the blood flow regulation field, but also very relevant to all with an interest in human brain mapping or astrocytic contribution to CNS pathologies.

The paper is in general well written and results clearly presented in the figures, though with a few flaws addressed as minor comments. The introduction and discussion refers to relevant papers and relate the findings in a proper way. With regards to the conclusions, they are in general sound though the authors might benefit from a more cautious evaluation of their methods in the discussion section. The genetic approaches to modification of cellular Ca²⁺ levels are very relevant and elegantly used but must be taken cautiously due to the changes they may cause in the complex and diverse regulation of intracellular Ca²⁺ levels.

Major comments:

- 1) *In the CalEx results the significant reduction in Ca²⁺ levels seem to be based on changes in the responses to 5sec. rather than the 30sec. stimulations. This may be due to a very large variance not evident from the way the average traces are represented in F and G, using SEM around the mean. Still, it is surprising, given the clear reduction in the average trace, that this cannot be reproduced in summarized values. Especially the AUC evaluations where the effect of CalEx expression on 30s stimulations is quite small and not statistically significant. Is it possible that the CalEx effect is primarily to delay the rise time in the astrocyte Ca²⁺ levels? And wouldn't the more precise description of the effect be that the Ca²⁺ rise, diminished/delayed by the CalEx, must be a recruiting factor of the second part of the biphasic dilation? This goes well in hand with the data presented in figure 3 and 4 as well.*

As explained in the opening, we have redone the CalEx experiment with an improved preparation and improved viral tools, and the results are much clearer and convincing than in our first submission. The new calcium traces from either the endfoot or the arbor, for 5sec or 30sec stim, clearly show a dramatically diminished amplitude of the calcium signal without “slowing” the calcium signal (control and CalEx calcium peaks occur at approximately the same time). This makes sense for how CalEx should work. The natural calcium elevation should still initiate as normal, but as calcium rises, the pump will work to lessen calcium concentration reached in the cytosol. We agree with the comment that the delayed astrocyte calcium signal comes on to amplify the second phase of vasodilation, but for the calcium to ramp up and there to be an effect on the vessel likely takes more than ~5-7sec. By this time, the initial response has already peaked. Our interpretation is that CalEx decreases the magnitude (but not necessarily slowing the timing) of the astrocyte calcium signal. We hope this satisfies the reviewer’s previous concern.

Regarding the older CalEx data, which is now removed, the discrepancy between the average traces and summary dot plot occurs based on how we decided to plot and analyze the data. We felt the most rigorous way was to show the average trace of the entire dataset with SEM, as well as plot the peak response from individual trials in the dot-plot summary data. This gives the reader two different ways to look at the data. The statistics are performed on the peak responses from the individual trials. We felt this was more appropriate than “picking” a common time point from the average trace which has some subjectivity. This is the case throughout the paper. We plot the average + SEM traces and show individual amplitudes or AUC in the summary data.

- 2) *With regards to the variance in the astrocytic Ca²⁺ signals several of the authors contributed to a study of the heterogeneity of the Ca²⁺ responses, showing that roughly 50% of the astrocyte endfeet surrounding PA responded to a whisker stimulation (3). How is that reflected in the averaged traces? And in the max peak and AUC Ca²⁺ data? Have the authors excluded Zero-responses events or included them in the averages? Were the occurrence of astrocytic PA endfeet Ca²⁺ responses evaluated, and did it change with the three tools applied to modify it (CalEx, Dread, AP5)?*

We included the data from every trial in the average trace data and in the summary dot plots (including zero responses), because we wanted to see the overall effect without picking traces from each dataset. Selecting and comparing ‘responders’ is possible if one applies criteria to each trace to determine if there was a ‘signal’ or not, such as if the trace rises 3 standard deviations above the mean within a certain time window after stimulation. We have used this trial-based selection technique previously to try and understand the variability in this system in the control condition, as the reviewer indicated. However, when asking if CalEx or Gq-DREADD influences arteriole diameter and calcium signals, looking at individual ‘responding’ traces can be misleading because one cannot easily disentangle changes in the amplitude with a change in failure rate. This is because as the amplitude decreases, the failure rate will also increase as more events drop below the criteria threshold. The reverse is also true if amplitude increases. Nevertheless, we decided to examine the *success vs failure* rate in the new control and CalEx experiments. Note, in the chronic window preparation using AAV to drive GCaMP6 expression, we see a greater success rate for detecting calcium signals in astrocytes (80% in the arbor and 65% in the endfoot to 5sec stim) compared to our previous experiments in an acute window using Rhod-2 or knock-in cre-lox animals. Notably, to 30sec stimulation the chance of detecting a calcium signal increases even more (93% arbor and 88% endfoot). As expected, we found that CalEx dramatically reduced the occurrence of “detected” calcium signals in all groups/conditions. This data can be seen in the new Fig. 2h,k.

- 3) *The authors use an air puff mediated whisker stimulation to initiate neurovascular response in the barrel cortex. Though it is mentioned in the methods section that it is given without blowing air in the face of the mouse, the reader would benefit from a brief assurance of the non-startling nature of this stimulation paradigm from the very beginning of the results section. This is important to underline given that air-puffs are used as startling stimulations which will induce a large Ca²⁺ response in astrocytes^{4,5}. There is an indirect affirmation of potentially startling effect of air puffs in the use of a “untrained body air puff” to startle the animals in supplemental figure 2. The paper would benefit from clearly addressing this issue, preferentially through a small investigation into the connection between these puff stimulations and pupil size or similar (6).*

We thank the reviewer for this interesting suggestion to track the pupil to get a better understanding of any startle-related effects in our data. As the reviewer knows, pupil diameter changes from startle/arousal and concomitantly changes with locomotion as well. To address this point, we have taken detailed measurements of pupil diameter in our new CalEx data set, while also tracking locomotion. But before we describe the new data, we attempted to address the issue of startle in our 2018 Neuron paper. A general startle response, such as puffing air in the face, yields a global elevation in astrocyte Ca²⁺ as well as vasodilation in both cortical hemispheres. We were able to show that examining astrocyte and arteriole responses in the ipsilateral barrel cortex to whisker stim yielded little astrocyte calcium signal nor vasodilation, suggesting startle was not a major component of what we were studying. In our previous submission of this paper, we showed analogous data comparing untrained vs trained air puff onto the body as the reviewer indicated. Nevertheless, when experimentally applying any sudden stimulus to a mouse (sound, light, air etc.), we think it is unlikely to have zero startle; thus, some subtle startle effects may be present in the data. To this end, our new pupil data, presented in Fig. 3, suggests this. For the best measurements, we trained a model of pupil

diameter across its horizontal and vertical axis and analysed the data using DeepLabCut. As a positive control, we found that intentionally startling the animal with untrained ipsilateral body puff for 30sec, produced a significant increase in pupil diameter compared to no stimulation at all. This response occurred alongside an immediate, relatively large increase in locomotion with some persisting locomotor bouts throughout the trial. When examining pupil diameter in response to whisker stimulation, we observed a trend to increasing pupil size, though the result was not statistically different from no stimulation ($p=0.066$) with a large number of trials compared ($N=16$). The pupil responses to whisker stim were associated with a transient increase in locomotion at the onset of the stimulation and at the offset (Fig. 3a,b). In summary, whisker stimulation produces a modest pupil dilation as a sign of a mild degree of startle/arousal, which cannot be easily separated from pupil dilation caused by locomotion itself in our passive treadmill experiments (McGinley et al., 2015). We have added a sentence to this effect on page 13, line 263-264 and page 24, line 441. Additionally, to test if any differences in startle could underly the difference we see in astrocyte CalEx on arteriole diameter, we confirmed that pupil responses were similar in CalEx and CalEx mutant control. We have added these new data to the results on page 13, line 266 and added a new figure (Figure 3).

- 4) *Just as a stimulation paradigm may influence the vascular and cellular responses through neuromodulators, locomotion will also influence the activity of the barrel cortex (7). The authors do not address this issue. If responses during locomotion is disregarded this should be clearly stated in the text. If not, it may have confounded the results and added the variance and timing of the vascular and astrocytic Ca²⁺ responses.*

As described above, we have added a detailed account of locomotion from every trial conducted in our new CalEx experiments. The results are consistent with the literature, while adding new knowledge. By capturing the surface of the Styrofoam ball at high frame rates with a camera, we measured relative locomotion. As described at the opening of this rebuttal, we found that overall running was not different between the two groups, suggesting that a difference in running cannot explain the reduction in the late phase of vasodilation we measure in astrocyte CalEx. This data has been added to the results on page 13, lines 233-253, with data shown in Fig. 3.

In Fig.4 the authors have chosen to present the effect of AP5 on vascular diameter in another set of animals than those used to evaluate the effect on astrocytic Ca²⁺. While this is not disqualifying it is puzzling, given that it appears as if the Ick-GCaM6f measurements were done in combination with Rhodamine D as a vascular marker. Why are the diameter changes not evaluated in the same mice as the Ca²⁺ changes in astrocytes following the AP5 based modification of the NMDAR activity? Finally, I am unconvinced that these experiments show that the effect is through astrocytic NMDAR alone, but the authors also acknowledge this in the discussion.

We needed to conduct the diameter measurements again in a separate set of experiments due to technical limitations. The diameter measurements were not optimal in the Ick-GCaMP6 experiments when tuned to 920nm, where Rhod-B-dextran is poorly excited. While we did acquire this data, we thought it had too poor signal-to-noise to provide accurate measurements, so we performed the AP5

test again in C57Bl/6 with intravascular Rhod-B-dextran and imaged at a shorter wavelength to have accurate measurements of diameter.

We agree about the location of the NMDARs. The effect could be through neuronal, endothelial or astrocytic NMDA receptors. All one can conclude from this experiment is that the entire late component of functional hyperemia depends on NMDA receptor activation, and if astrocytes contribute to the late component, there should be some NMDA-dependent involvement of astrocytes whether direct or indirect. Similar points remain in our discussion on page 24-25, line 450-462.

Minor issues:

Some of the data shown in Fig 1 seems unnecessary to convey the information. Why do we need section C to show the nature of the astrocytic Ca²⁺ when it is similar to and as clearly visible from the data presented in panel D? I am not certain the interest if the readers stretch to the sensitivity of Rhod2AM compared to the GDaMP6s as an indicator on astrocytic Ca²⁺ levels.

We agree with the Reviewer and have decided to remove the Rhod-2 data from the paper. We included it originally because the Rhod-2 signal follows the late phase of the dilation better than GCaMP. We suspect this is due to additional mitochondrial loading of the Rhod-2. However, seeing that we do not pursue the involvement of mitochondria in this work, we decided it was not necessary to include.

There seems to be a redundancy in the text referring the results from Fig2E. All text from line 169 “Again, peak...” to “stimulation (Fig. 2E).” in line 173 refer to what is already clear presented from line 164.

This is now fixed.

In a few places the large multi-figures could benefit from a more systematic organization. Examples: Fig.1 B “arteriole diameter” is written above the pictures in, but in C “astrocyte endfoot Ca²⁺” is written below the pictures in. In Fig.2 E F and G aligning the text headlines would give a less “jumpy” impression. Fig.4 F, two air puff capillaries in cartoon.

We have revamped all the figures and considered the reviewers comments to make them better organized and more systematic.

I find it peculiar that the author refers to the studies that show biphasic response curves as recent (line 64), since they are from 2011 and 2013 respectively.

Thank you for spotting this error, it is now changed.

On the figure it is written Rhodamine D as an abbreviation for Rhodamine-B-dextran. I find that to be suboptimal solution. Maybe Rhodamine B or Rhodamine-dex?.

Thank you for spotting this. All relevant figure panels now read “Rhod-B dextran”

Supplemental fig4 A is very difficult to read, especially when printed out on A4 format.

We were not sure why this was the case but hopefully our new supplemental figure section does not suffer from this problem.

Reviewer’s Reference List

1 Mishra, A. et al. Astrocytes mediate neurovascular signaling to capillary pericytes but not to arterioles. *Nat Neurosci* 19, 1619-1627, doi:10.1038/nn.4428 (2016).

2 Biesecker, K. R. et al. Glial Cell Calcium Signaling Mediates Capillary Regulation of Blood Flow in the Retina. *J Neurosci* 36, 9435-9445, doi:10.1523/JNEUROSCI.1782-16.2016 (2016).

3 Sharma, K., Gordon, G. R. J. & Tran, C. H. T. Heterogeneity of Sensory-Induced Astrocytic Ca(2+) Dynamics During Functional Hyperemia. *Frontiers in physiology* 11, 611884, doi:10.3389/fphys.2020.611884 (2020).

4 Ding, F. et al. alpha1-Adrenergic receptors mediate coordinated Ca2+ signaling of cortical astrocytes in awake, behaving mice. *Cell Calcium* 54, 387-394, doi:10.1016/j.ceca.2013.09.001 (2013).

5 Zuend, M. et al. Arousal-induced cortical activity triggers lactate release from astrocytes. *Nat Metab* 2, 179-191, doi:10.1038/s42255-020-0170-4 (2020).

6 Reimer, J. et al. Pupil fluctuations track rapid changes in adrenergic and cholinergic activity in cortex. *Nature communications* 7, 13289, doi:10.1038/ncomms13289 (2016).

7 Shimaoka, D., Harris, K. D. & Carandini, M. Effects of Arousal on Mouse Sensory Cortex Depend on Modality. *Cell reports* 25, 3230, doi:10.1016/j.celrep.2018.11.105 (2018).

Reviewer #3 (Remarks to the Author):

The manuscript by Institoris et al tackles an important question in the field of neurovascular coupling: what is the role of astrocytes in triggering or modulating vascular responses? Most studies have used correlations between blood flow changes and astrocyte signals to infer or discard any role for these cells. Initially, mostly somatic signals were measured for technical reasons, resulting in misleading interpretations. Then, monitoring of astrocyte processes revealed that astrocytes could potentially be involved in NVC whereas the persistence of functional hyperemia in IP3R2 KO animals questioned this hypothesis. Here, the authors used an elegant approach to silence astrocyte calcium and observed that it modulates NVC. I find the topic interesting, but several issues need to be addressed.

Major Comments:

1) In their introduction, the authors overstress that anesthetics alter astrocyte responses. The problem is widely known but more subtle, as it depends on the drug and concentration used. Data from anesthetized animals are certainly as relevant as those from the current study slice experiments. I therefore suggest that the authors introduce the topic with the paper by Schultz et al., which was the first to seriously correlate the activation of astrocytes with the stimulation intensity and the appearance of a delayed vascular response. The spatial distribution of the delayed response was also investigated by Chen et al. 2014. Finally, in the current study, all the experiments in awake mice are performed immediately after surgery, starting 30 min after the animals awoke. It is certain that recovery from surgery per se requires more than 30 mins as well as the complete recovery from isoflurane anesthesia (Shirey et al. 2015). The authors should therefore consider that their preparation is not fully normal, but this does not lower the value of their findings.

We have made some modifications to the introduction as suggested, starting with the Schulz paper (page 3, line 62-63) and now also citing the publication by Chen et al 2014. The text now reads:

“Under anesthesia, astrocyte calcium signals can be suppressed, and arteriole/CBF responses to sustained sensory stimulation consist of an initial rise followed by a plateau (Chen et al. 2014).”

After surgery and intravascular dye injection, the animal is transferred to the treadmill. It takes a fair amount of time to square the objective lens with the off-axis window. Then we take multiple z-stacks using a 16x 2P objective to map the anatomy of the window. Then we determine which penetrators respond to whisker stimulation and/or map the region with IOS imaging to see the hot spot of activation. Then we switch to a 40x higher resolution objective lens. During all this each animal gradually comes out of anesthesia and begins to move on the ball to which they are already trained. We do not collect any trial data until mice *appear* fully awake and running. The time from anesthesia cessation to collecting data is ~30min but our imaging sessions typically last 2-3 hours. Many times, multiple penetrating arteriole responses are measured, and we wait a minimum of 7min between 30sec stimulation trials and 2min between 5sec stimulation trials. Nevertheless, the point the reviewer makes is well taken. We cannot exclude residual effects from the surgery/anesthesia but most of the experiments are done this way, and our physiological responses are robust (30 sec stim = 25% dilation).

Importantly, we now add chronic window imaging data with our updated CalEx experiment, which in our minds is the most critical set of experiments in the paper. Here, mice received a fully sealed window with the dura mater left intact and the mice recovered for 4-6 weeks. Mice were still anesthetized with

isoflurane to inject vascular dye immediately before imaging, but only for 2-4min. We find the recovery from this brief exposure to anesthetic much faster than a long craniectomy surgery. We still found a reduction in the late component of functional hyperemia in CalEx compared to control virus in chronic windows similar to our acute window data we reported previously.

2) The authors mostly show average trace data from many animals. It emphasizes the reproducibility between animals but masks resting state fluctuations and the precise determination of onsets. For example, in Fig1C Ca²⁺ responses to 30 s stimulations seem to start almost immediately whereas in Fig1D a delay is observed, and the response lags the beginning of the astrocyte process signal. The authors quantify the latency as "the time from stimulation onset until 2 consecutive values exceed 3SD of the 2s baseline". It is probably too cautious, and I suspect that Ca²⁺ from processes would precede endfoot Ca²⁺ if the onset were determined with 2SD of the 2s baseline or using other approaches (the observation also applies to Fig4). Could the author check the hypothesis? Note that the conclusion of this result section (and Fig1) could be that the mechanisms underlying both early and late vascular responses start at the level of the SMCs and not astrocytes.

Perhaps the reviewer misread the summary Fig1E. In cyto-GCaMP6s mice, the astrocyte processes Ca²⁺ elevation does come before the endfoot, but both structures lag the dilation onset using our 3SD criteria. However, the reviewer's point about the technical and methodological differences in identifying the onset of the various signals prompted us to investigate the onset latency of vasodilation and astrocyte Ca²⁺ transients in a preparation that resembles the experiments where ultrafast (<1sec onset) astrocyte Ca²⁺ signals were detected (Lind et al., 2018 Glia and Stobart et al., 2018 Neuron). Here, using a chronically implanted cranial window over the dura (>4 weeks recovery) in cre-lox knock-in *Aldh1l1-Cre/ERT2* x R26-Lck-GCaMP6f mice (where astrocytes express the membrane-tethered fast GCaMP6f) we found that 47% of astrocyte process Ca²⁺ (but not endfoot Ca²⁺) begin prior to arteriole dilation, even when using the 3SD criteria for signal onset detection. At the same time, we also calculated the size of this early process Ca²⁺ signal and found that within the first sec of stimulation which precedes the peak of the initial vasodilation, the elevation of Ca²⁺ only reaches $\Delta F/F=3.6\pm 1.3\%$, which is significantly less than the size of the signals $\Delta F/F=22\pm 3.5\%$ between 3-5sec of stimulation ($P=0.0003$) as well as the overall process Ca²⁺ peak $\Delta F/F=46.8\pm 5.7\%$ ($P<0.0001$). This investigation was added to the text (on page 4 lines 95-105) and as a new supplemental figure (Suppl. Fig. 2).

We also guide the reviewer to scrutinize the kinetics of our traces from our improved CalEx dataset, where all the Ca²⁺ traces of the cytosolic GCaMP6f traces are very clear and show the same trend in the mutant control as in our acute experimental preparation in Fig. 1: first vessel, then processes, then endfoot. We do not believe it is necessary to pursue further detailed kinetic analysis as many other papers have done this already and our main thrust was to determine if lowering and clamping astrocyte free calcium *in vivo* has an impact on the arteriole response.

We agree with the possibility that the delayed astrocyte Ca²⁺ elevation may be in response to the vessel, thereby helping to amplify the vasodilation, as we attempted to show in our Tran et al. 2018 paper, but we cannot make any conclusions based on onset time data alone.

3) CalEx experiments:

-To clamp astrocyte Ca²⁺, the authors use AAVs. Suppl Fig2 shows that response latencies in both control and CalEx animals seem larger than in Rhod2 or Aldh1GCaMP6 animals. Could the authors quantify and/or discuss these differences?

We thank the reviewer for this comment and decided to quantify the different latencies in a chronic window preparation using cre-lox knock-in GCaMP6 animals, see above. The Rhod-2 data has been removed from the paper and the CalEx data has been updated. In the new CalEx dataset, the summary traces clearly show the sequence of the signals. We have not quantified and compared the onset latencies of astrocyte Ca²⁺ between CalEx and control, as the Ca²⁺ signals in the CalEx are so miniscule and rare that such comparison is meaningless.

3b) *-Fig2E shows that the effect of CalEx on average traces of the initial dilation peak differ for 5s and 30s stimulations. Why is this difference not observed in the plots (peak $\Delta D/D$)? There should be a better match between the plots and the traces shown, in particular as the authors conclude that CalEx acts primarily on the late component of functional hyperemia.*

We agree with the reviewer's assessment here on the data in our original submission, however, with the new and improved CalEx data, this point is no longer relevant.

4) In vitro experiments:

Could the authors clarify what stimulation protocols are used to trigger dilation in vitro. In Suppl Fig 4 it seems that different stimulation frequencies are used for 30s (Suppl Fig4, high frequency) and 5s stimulations (Suppl Fig5, theta burst). Given this assumption is right, why theta burst stimulation was not systematically? It would ease the comparison.

In the astrocyte patch clamp experiments, we wanted to consistently evoke but then eliminate the astrocyte Ca²⁺ signals with BAPTA, so we used a stimulation protocol (high intensity theta burst) that readily elicits astrocyte Ca²⁺ signals in control conditions. Here, we only changed the length of the stimulation. Both 5sec and 30sec theta burst stimulations evoked astrocyte Ca²⁺ signals but the length of stimulation influenced the calcium dependence of the vasodilation.

In the MSPPOH slice experiments, we wanted to contrast a stimulation paradigm that is sub-threshold to astrocyte Ca²⁺ activation (low intensity 5sec 20Hz: similar to Institoris et al. 2015 JCBFM) to a supra-threshold stimulation (theta burst) that we knew was effective at raising astrocyte free calcium. Albeit we changed two parameters at once across these two experiments, they demonstrate that if a stimulation does not elicit Ca²⁺ signals in astrocytes, then blocking EETs production has no effect on vasodilation. In contrast, if we apply a long stimulation that produces an astrocyte-mediated dilation (based on the patch data), then blocking EETs production is effective at reducing the arteriole response. From our astrocyte BAPTA experiment we also know that 5sec of theta burst-induced vasodilation does not depend on astrocyte calcium, and thus MSPPOH is also expected to have no effect on dilation using

this protocol. Though we did not specifically test 5sec theta burst in MSPPOH, the result can be reasonably inferred.

5) Chemogenetic activation:

-The effect of compound 21 on the late phase dilation is nice. As the effect on resting calcium recovers during the 1hr pretreatment, could the authors show if the dilation increase is matched by a Ca²⁺ response increase in astrocytes.

Unfortunately, we were unable to measure an enhanced astrocyte Ca²⁺ signal in the Gq-DREADD experiments. We had a technical limitation of suboptimal Rhod-2/AM loading in this experiment that was unexplained. While we were able to capture the large astrocyte Ca²⁺ signal in response to the initial C21 administration, the signal-to-noise was inadequate to capture endfoot Ca²⁺ elevations to whisker stimulation. These calcium trace data were thus inconclusive. We have added this limitation to page 15, line 304. We feel that the most important observations are clear: CNO arrives in the brain and activates astrocytes; 5sec dilations are not affected but 30sec dilations show a clear enhancement in the presence of C21.

6) NMDA receptors:

1 mM DLAP5 blocks Ca²⁺ responses in astrocytes and the delayed dilation. What is clearly missing is the effect on neurons as prolonged stimulation is likely to progressively activate neuronal NMDA channels. Could the authors control the extent to which the neuronal response is blocked by DL-AP5?

Same response to R1: For the topical application of the NMDA receptor antagonist AP5, in which we observed a complete elimination of the late phase of functional hyperemia, we agree with the reviewer that more experiments are necessary to better understand what is happening to neural activity. Specifically, it is important to know if AP5 abolishes the late component of neural activity during sustained sensory stimulation, because if true, then astrocytes need not be recruited in late functional hyperemia in an NMDA receptor-dependent manner. Thus, we have performed additional *in vivo* experiments to measure neural activity using Thy1-GCaMP6f mice, and test 5sec and 30sec whisker stimulations in the presence or absence of AP5. When measuring the bulk neuronal calcium signal directly around the arteriole (5000 micron²), we found that AP5 reduced the neuronal calcium signal to both 5sec and 30sec whisker stimulation. Notably, for 30sec stim there was a uniform reduction over the entire length of the signal, again showing no preferential effect on the late phase. This data is added to the results on page 20-21 lines 358-369 and appears in Fig 5h. Nevertheless, as this is a non-specific calcium signal and the degree of block of vasodilation by AP5 is more pronounced than astrocyte CalEx, we cannot exclude the possibility that at least part of the reduction in late phase of functional hyperemia in AP5 is attributed to reduced neural activity. This is now stated in the discussion on page 24, lines 454-457.

REVIEWERS' COMMENTS

Reviewer #1 (Remarks to the Author):

The authors fully addressed all my comments with new experimental data and submitted a thoroughly revised manuscript. The data presented show that the cerebrovascular responses to neuronal activation in the barrel cortex consist of two components – immediate, developing in response to stimuli of short duration; and sustained, recorded in response to longer sensory stimulation. The authors made two important conclusions: 1) the immediate neurovascular response is independent of astroglial Ca²⁺; 2) astrocytic Ca²⁺ signalling contributes to the late component of neurovascular coupling response, when the neuronal activity is sustained. The study is expertly performed, the data analysis is appropriate, the text of the manuscript is well-written, and the data are beautifully presented. This new study contributes in a significant manner to our understanding of the mechanisms of neurovascular coupling by clarifying the role played by Ca²⁺ signals in astrocytes. I would like to make two additional points which the authors may wish to discuss in the paper and/or consider in their future work:

1. In the first version of the manuscript the authors reported the profiles of astrocytic Ca²⁺ responses recorded using Rhod-2 and GCaMP6s that were very different. Rhod-2 recorded Ca²⁺ elevations that were largely sustained during the 30-s period of stimulation and returned to baseline after the period of stimulation. Signals reported by GCaMP6 returned to the baseline well before the termination of the stimulus (Figs 1f, 5b,d, but not Fig 2i,l, see below). The authors have now excluded Rhod-2 data from the revised submission, and I would probably suggest the authors to include these results, perhaps in the Supplementary Material. GCaMP sensors are based on GFP and GFP fluorescence is known to be sensitive to pH (PMC: 1299504). What if Rhod-2, in fact, accurately reports sustained changes in Ca²⁺, while decreases in GCaMP6 fluorescence during longer periods of stimulation simply reflect the activity-dependent intracellular acidification of astrocytes, resulting in quenching of GCaMP fluorescence?

2. I am sure the authors noted that Ca²⁺ responses in astrocyte endfeet and processes recorded in the preparations with chronic, fully sealed cranial windows are very different to that recorded in preparations with acutely implanted cranial windows or perforated coverglass windows (compare Fig 2i,l and Figs 1f, 5b, 5d). The responses recorded in animals with fully sealed cranial windows look remarkably similar to Ca²⁺ responses induced in astrocytes by mechanical stimulation, with two response peaks on stimulus onset and offset (see Fig 2 in PMID: 31919423). These data suggest that astroglial Ca²⁺ responses recorded in the imaging experiments of this type may critically dependent on the integrity of (and the pressure within) the intracranial space.

Reviewer #2 (Remarks to the Author):

The revised version of the manuscript “Astrocytes amplify neurovascular coupling to sustained activation of neocortex in awake mice”, by Institoris et al., is much approved by the substantial effort put forth by the authors in addressing the concerns of the initial revision. The result is a strengthened manuscript that convincingly show an astrocytic contribution to the prolongation of the hemodynamic response to sensory stimulation in awake mice. That a noradrenergic-/arousal components to this response cannot be ruled out muddies the picture, but this contribution is physiological relevant and it does not go uncommented by the authors. Thus, I support the publication of this manuscript.

Minor:

Line 354 for the loss ...of?...evoked astrocyte calcium

Figure 5 is still a little bit chaotic with regards to the position/presence of T=_,PA=_, N=_____ data.

Reviewer #3 (Remarks to the Author):

In their revised article, the authors have added a lot of work to answer the questions raised by all the reviewers. I am particularly pleased with the improvement in data quality, which now untangles several issues. The work is now publishable as is.

Final reviewer comments

Reviewer 1

1. In the first version of the manuscript the authors reported the profiles of astrocytic Ca²⁺ responses recorded using Rhod-2 and GCaMP6s that were very different. Rhod-2 recorded Ca²⁺ elevations that were largely sustained during the 30-s period of stimulation and returned to baseline after the period of stimulation. Signals reported by GCaMP6 returned to the baseline well before the termination of the stimulus (Figs 1f, 5b,d, but not Fig 2i,l, see below). The authors have now excluded Rhod-2 data from the revised submission, and I would probably suggest the authors to include these results, perhaps in the Supplementary Material. GCaMP sensors are based on GFP and GFP fluorescence is known to be sensitive to pH (PMC: 1299504). What if Rhod-2, in fact, accurately reports sustained changes in Ca²⁺, while decreases in GCaMP6 fluorescence during longer periods of stimulation simply reflect the activity-dependent intracellular acidification of astrocytes, resulting in quenching of GCaMP fluorescence?

We have added back the astrocyte Rhod-2 calcium data as the reviewer suggested. This is now presented in supplementary figure 3. We also added a short discussion section on this, speculating as the reviewer did on intracellular acidification or mitochondrial loading of Rhod-2. Specifically, we say:

“Notably, the profiles of astrocytic Ca²⁺ responses recorded using Rhod-2/AM and GCaMP6 were different. Rhod-2/AM recorded Ca²⁺ elevations were largely sustained during 30sec stimulation and returned to baseline only after stimulation. Signals reported by GCaMP6 recovered well before the termination of the stimulus. It is possible that Rhod-2/AM reported cytosolic Ca²⁺ changes are more accurate, since GFP-based GCaMP sensors are sensitive to pH⁸² and activity-dependent acidification of astrocytes may have led to quenching of GCaMP fluorescence during the late phase of 30sec stimulation. Alternatively, the difference could be explained by the preferential loading of Rhod-2/AM into mitochondria, an organelle in which GCaMP is excluded. Future experiments should explore whether astrocytic mitochondrial Ca²⁺ dynamics contribute to sustained functional hyperemia.”

2. I am sure the authors noted that Ca²⁺ responses in astrocyte endfeet and processes recorded in the preparations with chronic, fully sealed cranial windows are very different to that recorded in preparations with acutely implanted cranial windows or perforated coverglass windows (compare Fig 2i,l and Figs 1f, 5b, 5d). The responses recorded in animals with fully sealed cranial windows look remarkably similar to Ca²⁺ responses induced in astrocytes by mechanical stimulation, with two response peaks on stimulus onset and offset (see Fig 2 in PMID: 31919423). These data suggest that astroglial Ca²⁺ responses recorded in the imaging experiments of this type may critically dependent on the integrity of (and the pressure within) the intracranial space.

We thank the reviewer for this point and indeed noticed the difference. As its an important point and our paper now contains of mixture of chronic window experiments and acute window experiments, we have added a short discussion on this. We say that:

“We also acknowledge that astrocyte endfeet and process Ca²⁺ signals recorded in the preparations with chronic, fully sealed cranial windows are different to that recorded in preparations with acutely

implanted, sealed or perforated cranial windows. Astrocyte Ca²⁺ responses recorded through fully sealed cranial windows additionally revealed elevations that were time locked to the onset and offset of whisker stimulation. While this may reflect responses to onset and offset neurons in the barrel cortex⁴⁸, they could also reflect astrocyte mechanosensing⁸³ from the blood flow response itself^{11,84}, which would not be observed if the integrity of the intracranial pressure is compromised.”

Reviewer 2

The revised version of the manuscript “Astrocytes amplify neurovascular coupling to sustained activation of neocortex in awake mice”, by Institoris et al., is much approved by the substantial effort put forth by the authors in addressing the concerns of the initial revision. The result is a strengthened manuscript that convincingly show an astrocytic contribution to the prolongation of the hemodynamic response to sensory stimulation in awake mice. That a noradrenergic-/arousal components to this response cannot be ruled out muddies the picture, but this contribution is physiological relevant and it does not go uncommented by the authors. Thus, I support the publication of this manuscript.

Minor:

Line 354 for the loss ...of?...evoked astrocyte calcium

This is now fixed

Figure 5 is still a little bit chaotic with regards to the position/presence of T=, PA=, N= data.

We simplified Figure 5 by moving the MSPPOH data to a new Fig 6 and tried to reposition the labels to make things clearer.

Reviewer 3

In their revised article, the authors have added a lot of work to answer the questions raised by all the reviewers. I am particularly pleased with the improvement in data quality, which now untangles several issues. The work is now publishable as is.

No comments to address